# *KCNQ2* Selectivity Filter Mutations Cause Kv7.2 M-Current Dysfunction and Configuration Changes Manifesting as Epileptic Encephalopathies and Autistic Spectrum Disorders

**DOI:** 10.3390/cells11050894

**Published:** 2022-03-05

**Authors:** Inn-Chi Lee, Jiann-Jou Yang, Ying-Ming Liou, Swee-Hee Wong

**Affiliations:** 1Division and Pediatric Neurology, Department of Pediatrics, Chung Shan Medical University Hospital, Taichung 40201, Taiwan; 2Institute of Medicine, School of Medicine, Chung Shan Medical University, Taichung 40201, Taiwan; 3Genetics Laboratory and Department of Biomedical Sciences, Chung Shan Medical University, Taichung 40201, Taiwan; jiannjou@csmu.edu.tw (J.-J.Y.); a7710355@yahoo.com.tw (S.-H.W.); 4Department of Life Sciences, National Chung-Hsing University, Taichung 40227, Taiwan; ymlion@dragon.nchu.edu.tw; 5The iEGG and Animal Biotechnology Center, Rong Hsing Research Center for Translational Medicine, Natinal Chung Hsing University, Taichung 40227, Taiwan

**Keywords:** whole-cell patch-clamp analysis, ASD, epileptic encephalopathy, children, *KCNQ2*

## Abstract

*KCNQ2* mutations can cause benign familial neonatal convulsions (BFNCs), epileptic encephalopathy (EE), and mild-to-profound neurodevelopmental disabilities. Mutations in the *KCNQ2* selectivity filter (SF) are critical to neurodevelopmental outcomes. Three patients with neonatal EE carry de novo heterozygous *KCNQ2* p.Thr287Ile, p.Gly281Glu and p.Pro285Thr, and all are followed-up in our clinics. Whole-cell patch-clamp analysis with transfected mutations was performed. The Kv7.2 in three mutations demonstrated significant current changes in the homomeric-transfected cells. The conduction curves for V_1/2_, the K slope, and currents in 3 mutations were lower than those for the wild type (WT). The p.Gly281Glu had a worse conductance than the p.Thr287Ile and p.Pro285Thr, the patient compatible with p.Gly281Glu had a worse clinical outcome than patients with p.Thr287Ile and p.Pro285Thr. The p.Gly281Glu had more amino acid weight changes than the p.Gly281Glu and p.Pro285Thr. Among 5 BFNCs and 23 EE from mutations in the SF, the greater weight of the mutated protein compared with that of the WT was presumed to cause an obstacle to pore size, which is one of the most important factors in the phenotype and outcome. For the 35 mutations in the SF domain, using changes in amino acid weight between the WT and the *KCNQ2* mutations to predict EE resulted in 80.0% sensitivity and 80% specificity, a positive prediction rate of 96.0%, and a negative prediction rate of 40.0% (*p* = 0.006, χ^2^ (1, *n* = 35) = 7.56; odds ratio 16.0, 95% confidence interval, 1.50 to 170.63). The findings suggest that p.Thr287Ile, p.Gly281Glu and p.Pro285Thr are pathogenic to *KCNQ2* EE. In mutations in SF, a mutated protein heavier than the WT is a factor in the Kv7.2 current and outcome.

## 1. Introduction

*KCNQ2* (OMIM 602235)-associated seizures typically occur in the first week from birth and can contribute to benign familial neonatal convulsions (BFNCs), benign familial infantile seizures [1,2,3,4,5], and neonatal-onset epileptic encephalopathy (EE) [6,7,8]. *KCNQ2* BFNCs are generally predicted to follow a benign course and are expected to have unremarkable outcomes. The majority of neonatal-onset EE mutations are de novo; they are rarely mosaic. Patients with *KCNQ2* EE present with severe seizures that often remit as patients become older; however, such patients experience poor neurodevelopmental outcomes in terms of cognition, motor skills, and language, and they may also exhibit autistic spectrum disorder (ASD). This condition is called developmental EE [9,10]. Neurodevelopmental outcomes can be determined by genotype; the related functional consequences result from mutations caused by M-current changes, prolonged seizure duration, and environmental factors. Among these factors, genotype is the most critical in determining outcomes [11,12,13,14].

*KCNQ* channels, which include a voltage sensor in S1–S4 and S5–S6 as well as a loop between S5 and S6 that creates an ion channel pore, a cytoplasmic N-terminal, and a long C-terminal region with complex functions in interactions among syntaxin, phosphatidylinositol 4, 5-bisphosphate, ankyrin-G, Syn-1A, and A-kinase anchoring proteins [2,5,15,16,17,18,19,20]. The S5 and S6 segments, along with an intervening reentrant loop (P-loop domain), form the pore region. Mechanisms that govern the expression of Kv7.2 include the network of interactions of the pore region with the selectivity filter (SF) and the S6, which are responsible for the Kv7.2 current and control of the KCNQ2 protein on the plasma membrane by the C-terminus distal part for channel trafficking and assembly. The C-terminus includes two helical domains (helices A and B) for channel modulation through interaction with calmodulin [21,22]. Helix A contains the consensus CaM-binding IQ motif, and helix B mediates Ca^2+^-dependent CaM binding [23]. All *KCNQ* channel members have large C-termini that may form “receptorsomes” or “channelosomes” for incorporation with multiple signaling pathways [24]. Mutations in the *KCNQ2* gene can cause haploinsufficiency and a severe dominant-negative effect due to a loss of function and gain of function. Loss of function explains the majority of cases of neonatal-onset EE [25,26,27,28], but a gain in function is presumed to be the mechanism behind several mutations [25,28,29,30,31,32]. The mutations located on the SF of the pore domain significantly impact the Kv7.2 and cause severe functional consequences. The SF located between S5 and S6 is highly conserved and controls K^+^ permeability and selectivity [33,34,35,36,37]. The SF is critical for maintaining K^+^ permeability and selectivity; its amino acid position in *KCNQ2* is from protein 253 to 291.

*KCNQ2* EE mostly manifests as ASD and poor neurodevelopmental behaviors such as language delay, stereotypical behavior, and seizures. In patients with *KCNQ2*-related BFNC, the reported neurodevelopmental outcomes are healthy neurodevelopment, mild intellectual disability, and attention deficit disorder. In one case, *KCNQ2* EE caused hyperkinetic movement of the limbs beyond age 4 weeks [38]. Neonatal seizures associated with severe neonatal myoclonus such as dyskinesia due to a familial *KCNQ2* gene mutation was reported [39]. However, the manifestations beyond neonatal age was rarely reported.

The SF in *KCNQ2* is from protein 253 to 291 (Figure 1). We found 35 mutations [5 caused BFNCs (14.3%) and 30 caused EE (85.7%)] according to a comprehensive literature review of the literature. All neurodevelopmental outcomes related to BFNCs were favorable. Among patients with mutations causing EE, the outcomes were poorer than the outcomes of those with mutations causing BFNCs and varied from moderate or severe to profound developmental disabilities and early mortality.

The precise genotype–phenotype correlation in *KCNQ2*-related epilepsy is not fully understood. We demonstrated that patients with *KCNQ2* EE exhibited dyskinetic movement disorders and ASD after infantile age and determined that three *KCNQ2* variants in the SF of the pore domain caused functional current changes in HEK293 cells. Second, to predict the consequences of mutated proteins in the *KCNQ2* SF, we hypothesized that mutations correlated with phenotype and neurodevelopmental outcomes were also correlated with changes in the mutated protein of the SF. We analyzed various mutations in the SF to predict structural changes at the molecular level.

## 2. Materials

### Participants

Two hundred twenty-six patients were enrolled from 2015–2020, and met the criteria for “childhood epilepsy without an identified cause” ([1] age of first seizure less than 18 years old and [2] at least one magnetic resonance image (MRI) with no detectable seizure-related lesions) after excluding patients with the obvious MRI findings indicating lesional epilepsies. Ten (4.4%) patients (5 boys; 5 girls) had *KCNQ2* mutations: one each for c.860 C > T (p.Thr287Ile); c.740 C > T (p.Ser247Leu); c.842G > A (p.Gly281Glu); c.853 C > A (p.Pro285Thr); c.1294 C > T (p.Arg432Cys); c.1342C > T (Arg448Ter); c.1627 G > A (p.Val543Met); c.1741 C > T (p.Arg581Ter) and a splicing mutation: c.387 + 1 G > T. The *KCNQ2* sequencing data were compared with the GenBank reference sequences and the version numbers of the *KCNQ2* gene (NM_172107.3).

Three (30%) out of ten mutations located in the SF of the KCNQ2 protein were p.Thr287Ile, p.Gly281Glu and p.Pro285Thr mutations, and presented with neonatal-onset EE, movement disorders and ASD. We transfected these variants (p.Thr287Ile, p.Gly281Glu and p.Pro285Thr) into HEK293 cells to investigate Kv7.2 current changes and the *KCNQ2* protein expression in HEK293 cell membranes.

## 3. Methods

### 3.1. Computational Protein Analysis of SF Mutations and Their Phenotypes

A molecular model of *KCNQ2* channel proteins (NM_004518) was generated using the Phyre2 tool (Protein Homology/analogY Recognition Engine V 2.0, Imperial College, London) and the NP_004509.2 protein sequence. This tool can be used to conduct protein modeling, prediction, and analysis based on the CryoEM structure of the Xenopus *KCNQ1* channel [40]. The predicted 3D model of the *KCNQ2* channel protein (c5vmsA_.1.pdb) was then used along with SPDBV and PyMOL to analyze the structural differences between wild type (WT) and mutant cells. The 3D structure was predicted through homology modeling using the Phyre2 database and was validated using SPDBV. Common characteristics of the predicted protein, such as molecular weight, isoelectric point, amino acid composition, and the aliphatic and instability indexes, were assessed using the ProtParam tool. We used the Human Gene Mutation Database (HGMD) (http://www.hgmd.cf.ac.uk/ac/index.php, accessed on March 2021) and National Center for Biotechnology Information (NCBI) ClinVar (https://www.ncbi.nlm.nih.gov/clinvar, accessed on March 2021) database to assess SF mutations in the pore domain. The WT and mutated protein characteristics were then analyzed.

### 3.2. Mutations of KCNQ2 in Corresponding to KCNQ2 Functional Domains and Phenotypes

We reviewed the database in HGMD and NCBI and confirmed the pathogenic characters of the mutations to review the corresponding literatures. We collected the missense mutations of KCNQ2. The phenotypes were classified to BFNC and KCNQ2 EE.

### 3.3. Expression in HEK293 Cells

HEK293 cells were maintained in Dulbecco’s modified Eagle’s medium (DMEM) (Biowhittaker, Walkersville, MD, USA) supplemented with 10% fetal bovine serum (FBS), penicillin (100 U/mL), streptomycin (100 U/mL), and 2 mM L-glutamine (Lonza, Walkersville, MD, USA). Mutations in *KCNQ2* were induced using a kit (QuickChange; Stratagene, La Jolla, CA, USA) and verified using sequencing [41].

### 3.4. Transfecting Variants to HEK293 Cells

HEK293 cell cultures were maintained at 37 °C in a humidified 5% CO_2_ incubator. The vectors pLEGFP and pTaqRFP, which contained DNA fragments encoding both wild-type and mutant *KCNQ2*, were transfected to HEK293 cells using a reagent (lipofectamine; Thermo Fisher Scientific: Invitrogen, Carlsbad, CA, USA). *KCNQ2* mutations were induced using a kit (QuickChange; Stratagene, La Jolla, CA, USA) and verified using sequencing. *KCNQ2* WT and mutant *KCNQ2* alleles were transfected into HEK293 cells in homomeric mutants (2 μg) and heteromeric *KCNQ2*+ mutants (1 μg + 1 μg), respectively. After transfection with heteromeric *KCNQ2* WT and its variants and WT *KCNQ3* (0.5 μg:0.5 μg:1 μg), the DNA ratio mimicked the genetic balance.

### 3.5. Whole-Cell Patch-Clamp Analysis

For electrophysiological analysis, HEK293 cells were washed in modified Tyrode’s solution containing 125 mM NaCl, 5.4 mM KCl, 1.8 mM CaCl_2_, 1 mM MgCl_2_, 6 mM glucose, and 6 mM HEPES (pH 7.4). Patch pipettes had a resistance of 3–4 Ω when filled with a solution containing 125 mM potassium gluconate, 10 mM KCl, 5 mM HEPES, 5 mM EGTA, 2 mM MgC_l2_, 0.6 mM CaCl_2_, and 4 mM adenosine 5′-triphosphate disodium salt hydrate (Na2ATP; pH 7.2). *KCNQ2* mutations were created using a QuickChange kit (Stratagene, La Jolla, CA, USA) and verified through sequencing [41]. To measure the voltage dependence of activation, the cells were clamped using 3-s conditioning voltage pulses to potentials between −80 mV and +40 mV in 10-mV increments from a holding potential of −80 mV. Data acquisition and analysis were using analysis software (Clampex 10.0; Molecular Devices, Sunnyvale, CA, USA). The data were then fitted to a Boltzmann distribution of the following form: *G**/Gmax* = 1/(1 + *exp*[(*V* − *V*½)/*dx*]). Cell capacitance was obtained by reading the settings for the whole-cell input capacitance neutralization directly from the amplifier [42]. *KCNQ2* mutation variants and *KCNQ2* WT were transfected into HEK293 cells to determine the functional changes resulting in *c**onductance*–current *curve* changes [14,43].

### 3.6. Cytoplasmic and Membranous Protein Separation

Three wells were covered with HEK293 cells (6 × 10^6^) in a 10-cm cell culture dish. The cells were washed two or three times with phosphate buffered saline (PBS) containing 4 g of NaCl, 0.1 g of KCl, 0.72 g of Na_2_HPO_4_, 0.13 g of KH_2_PO_4_, with an adjusted pH of 7.4. The cells were then added to wells containing sucrose in a homogeneous solution (40 mM of Tris-HCl [pH = 7.4], 0.34 M sucrose, 10 mM EDTA, 1 mM MgSO_4_), and where 1 mL of 1-mM phenylmethyl sulfonyl fluoride (PMSF) was then added. The cell mixture was placed on ice and sonicated three times for 2 min each (intensity: 30). It was then slowly added to a mixture of 1.5 mL of 50% sucrose in a centrifuge tube (40 mM Tris-HCl [pH 7.4], 50% sucrose, 10 mM ethylenediaminetetraacetic acid (EDTA), 1 mM MgSO_4_, 2 mM NaN_4_) and 0.75 mL of 20% sucrose (40 mM of Tris-HCl [pH 7.4], 20% sucrose, 10 mM EDTA, 1 mM MgSO_4_, 2 mM NaN_4_), and completely shook cell homogeneously. The solution was then centrifuged in an ultra–high-speed rotor (55-Ti; Beckman Coulter Taiwan, Taipei) at 4 °C and 26,200 rpm for 90 min. After the solution had been centrifuged, the cellular proteins rose to the top of the liquid.

### 3.7. Western Blotting

Samples were diluted to at least 1:5 with sample buffer, heated at 95 °C for 5 min, and then stored at 4 °C until use. The gel was run at 80 V for 10 min and then at 130 V for 3 h. To prepare for Western blotting, the polyvinylidene difluoride (PVDF) membrane (Millipore) was soaked in methanol for 1 min and then placed in the “sandwich” chamber with 2 fiber pads and 2 filter papers that absorbed the old transfer buffer. The “sandwich” was transferred for 1.5 h at 100 V at 4 °C. The membrane was then shaken in 5% nonfat dry milk in PBS for 1 h in a shaker at room temperature; it was then incubated overnight with a primary anti-KCNQ2 antibody (1:200; Thermo Fisher) in 1% milk at 4 °C in a shaker. The next day, after it had been washed with PBST (phosphate buffer saline + Tween20) four times for 10 min each time, the membrane was incubated with a secondary antibody (antirabbit) (1:3000; Gentex) in 1% milk prepared with PBS for approximately 1 h at room temperature. It was then rinsed with PBST four times for 10 min each time and analyzed using a Western blotting detection kit (Advansta, Menlo Park, CA, USA). An anti-GAPDH antibody was used as the internal control and anti-pan cadherin was used as a cell membrane marker.

### 3.8. Ethics

The Chung Shan Medical University Hospital’s Institutional Review Board provided the ethical approval for the study (IRB #: CS2-14003). Written informed consents were obtained from parents of all three patients.

### 3.9. Statistics

Significant differences between groups were evaluated using an independent *t* test to compare wild types and mutants. The chi-squared test was used to differentiate categorical variables, and the Fisher exact test was used when sample sizes were small. The odds ratio (OR) was calculated by dividing the odds of the first group by the odds of the second group, and OR represents the association between a variable and an outcome. Significance was set at *p* < 0.05. The exact *p* values are expressed, unless *p* is < 0.001. All statistical tests were carried out using SPSS (version 14.0; SPSS Institute, Chicago, IL, USA).

## 4. Results

Three patients with mutations in SF presented as *KCNQ2* EE and ASD. To determine that three *KCNQ2* variants in the SF cause functional current changes in HEK293 cells and to predict the consequences of mutated proteins in the *KCNQ2* SF, we analyzed the functional currents in the mutations in the SF, and predicted the structural changes at the molecular level by computational protein analysis.

### 4.1. Clinical Presentations in 3 Patients with KCNQ2 Mutations in SF Domain

Patient 1 carried de novo p.Thr287Ile (uncertain significance according to NCBI ClinVar databases) and presented with neonatal seizures since day 3 of life. He first received intravenous phenobarbital, which could not control his seizures. The addition of oxcarbazepine (OXC) and topiramate controlled his seizures. After 3 months, he was prescribed OXC and topiramate for seizure control. He could not walk, and he had a severe cognitive disability without development of any language at 3 years old. After age 3 years, his parents found that he had repeated dyskinetic movements while awakening. The stereotyped episodes could occur up to 30 times during night and day without external stimulation. Related movements could also occur when the patient played with his parents. Antiepileptic drugs such as levetiracetam and topiramate did not affect these movements. An electroencephalogram (EEG) monitor showed no paroxysmal activity when the movements occurred, proving that it was not a seizure (Table 1).

Patient 2 was aged 4 years and carried a de novo p.Gly281Glu mutation (likely pathogenic according to NCBI ClinVar). She presented with seizures since day 2 of life, and her condition was not responsive to intravenous phenobarbital and phenytoin. Numerous drugs were administered, including oxycarbamazepine (30 mg/kg/day). During the period of ictal seizures, the patient exhibited stereotypical right-hand tonic seizures during both sleeping and waking states. The seizures occurred up to five times per day. The ictal EEGs revealed rhythmic delta waves in the left hemisphere and were associated with right-hand tonic movements. At age 4 years, the patient had spontaneous hyperkinetic behavior without external stimulation and still had seizures on a weekly basis (Table 1).

Patient 3 with the de novo p.Pro285Thr mutation, had frequent neonatal seizures, and apnea. Her EEG showed burst-suppression. She had neonatal seizures and was treated with multiple antiepileptic drugs. The seizures abated 2 months after she had been treated with oxcarbazepine. The seizures became less frequent after she turned 2 months old, but she had a severe cognitive disability and no language development at 4 years of age (Table 1).

### 4.2. Electrophysiological Properties of p.Thr287Ile in KCNQ2 Mutations

We analyzed the variant (p.Thr287Ile) in which the cell M-current was affected after the transfection of homomeric and heteromeric variants (Figure 2A(a–c) and Table 2). In HEK293 homomeric-transfected variants, the p.Thr287Ile cells expressed significantly lower currents (Appendix A) than did the *KCNQ2* WT when they were transfected with *KCNQ2* (Figure 2A(d)); a normalized current was significantly lower in homomeric p.Thr287Ile after −30 to −20 mV and 0 mV episodes of conditional stimulation in HEK293 cells (Figure 2A(e)). The homomeric currents were lower in p.Thr287Ile (481.8 ± 56.9 mV; *n* = 10) than the currents in the *KCNQ2* WT (579.8 ± 46.0 mV; *n* = 22) (Table 2 and Figure 2A(d)). The conduction curves for V_1/2_ is right shifting 4.7 mV (−12.2 ± 1.9 mV in homomeric p.Thr287Ile versus −16.9 ± 2.0 mV in *KCNQ2* WT), and K (slope) [8.3 ± 1.3 (mV/e-fold) in homomeric p.Thr287Ile versus 9.5 ± 2.2 (mV/e-fold) in *KCNQ2* WT] were lower than those for the WT (Table 2). The conduction curve exhibited a significant impairment in the homomeric p.Thr287Ile channel.

In heteromerically transfected *KCNQ2* WT and p.Thr287Ilev (1μg:1 μg; *n* = 10), amplitudes were significantly lower in p.Thr287Ile (Appendix A) at a conditional voltage of 10 mV to +40 mV (Figure 2A(d)); the normalized current was lower in p.Thr287Ile + *KCNQ2* WT than did the *KCNQ2* WT in HEK293 cells (Figure 2A(e)). The p.Thr287Ile + *KCNQ2* WT cells expressed lower (*t* (30) = 1.99, *p* = 0.056) currents (519.4 ± 52.0 mV, *n* = 10) than did the *KCNQ2* WT (579.8 ± 46.0 mV; *n* = 22) (Table 2). After transfection with *KCNQ2* WT + *KCNQ 3* WT (1μg:1μg; *n* = 10) (Figure 2B(a)) and heteromeric *KCNQ2* WT + p.Thr287Ile + WT *KCNQ3* (0.5 μg:0.5 μg:1 μg; *n* = 10) (Figure 2B(b)), the DNA ratio of which mimicked the genetic balance, amplitudes were still lower in p.Thr287Ile at 0 mV to +40 mV than in the *KCNQ2* WT + *KCNQ 3* WT (Figure 2B(c)) (Appendix A); the normalized current was lower in p.Thr287Ile (Figure 2B(d)). At a +40 mV conditional stimulation, the currents in *KCNQ2* WT and *KCNQ2* WT + *KCNQ3* WT are superior to homomeric p.Thr287Ile, heteromeric p.Thr287Ile + *KCNQ2* WT, and *KCNQ2* WT + p.Thr287Ile + *KCNQ3* WT, correspondingly (Figure 2C(a)) (Appendix A). The normalized currents (G/Gmax) are different among channels, and significantly (*t* (30) = 2.39, *p =* 0.027) better in *KCNQ2* WT versus homomeric p.Thr287Ile at a −30 mV voltage (Figure 2C(b)). In the conduction curve in p.Thr287Ile + *KCNQ2* WT, the V_1/2_ was right shifting 3 mV with a lower slope K value [8.7 ± 1.1 (mV/e-fold) in p.Thr287Ile + *KCNQ2* WT versus 9.5 ± 2.2 (mV/e-fold) in *KCNQ2* WT] (Table 2). In the *KCNQ2* WT + p.Thr287Ile + *KCNQ3* WT, the conduction curve was closer to the curve of *KCNQ2* WT + *KCNQ 3* WT, but still had a worse conductance–current curve compared with *KCNQ2* WT + *KCNQ 3* WT.

### 4.3. Electrophysiological Properties of p.Gly281Glu in KCNQ2 Mutations

The homomeric p.Gly281Glu HEK293 cells expressed significantly lower currents (pA) than did the *KCNQ2* WT cells from −10 to +40 mV (Figure 3A(d)) (Appendix A). The normalized currents were significantly lower in homomeric p.Gly281Glu cells from −30 to −10 mV stimulation than in *KCNQ2* WT cells (Figure 3A(e)). In homomeric-transfected variants (*n* = 8), V_1/2_ was −10.9 ± 1.8 mV; this value was right shifted 6 mV from the V_1/2_ (−16.9 ± 2.0 mV) in *KCNQ2* WT. The slope (K) was lower [8.0 ±1.2 (mV/e-fold); *t*(28) = 2.27, *p =* 0.031] than that of the *KCNQ2* WT (Table 2). The p.Gly281Glu ratio of currents to *KCNQ2* WT was 77.2%. In heteromeric-transfected *KCNQ2* WT cells and variants (1 μg:1 μg) (*n* = 8), the conductance–current curves for *KCNQ2* WT + p.Gly281Glu revealed smaller increases in the current than in homomeric p.Gly281Glu (Figure 3A(d); Table 2). After transfection with the heteromeric *KCNQ2* WT + p.Gly281Glu + *KCNQ3* WT (0.5 μg:0.5 μg:1 μg) (*n* = 8), the current amplitudes remained lower in the *KCNQ2* WT + p.Gly281Glu + *KCNQ3* WT cells than in the *KCNQ2* WT + *KCNQ3* WT cells (Figure 3B(c,d)). At a +40 mV conditional stimulation, the currents in *KCNQ2* WT were superior to homomeric p.Gly281Glu (1534.5 ± 141.9 versus 1109.4 ± 112.6, pA/pF) (t(30) = 8.76, *p* < 0.001) and to heteromeric p. Gly281Glu + *KCNQ2* WT (1534.5 ± 141.9 versus 1323.1 ± 81.0, pA/pF) (t(30) = 4.18, *p* < 0.001); *KCNQ2* WT + *KCNQ3* WT were superior to *KCNQ2* WT + p. Gly281Glu + *KCNQ3* WT (2474.3 ± 395.3 versus 2120.0 ± 65.9, pA/pF) (t(18) = 2.49, *p* = 0.024), correspondingly (Figure 3C(a)) (Appendix A). The value of G/Gmax was significantly low at a conditional voltage of −30 mV in homomeric p.Gly281Glu [t(28) = 2.48, *p* = 0.024] and in heteromeric *KCNQ2* WT + p. Gly281Glu + *KCNQ3* WT [t(16) = 2.81, *p* = 0.012] compared with WT, correspondingly (Figure 3C(b)). The cell currents in homomeric p.Gly281Glu, hetromeric p. Gly281Glu + *KCNQ2* WT, and *KCNQ2* WT + p. Gly281Glu + *KCNQ3* WT were impaired to be lower than in the WT.

### 4.4. Conductance–Current *Curves* in p.Thr287Ile, p.Gly281Glu, p.Pro285Thr and KCNQ2 WT

The p.Pro285Thr conduction curve is exhibited in Figure 4. Among the 3 mutations, the V_1/2_ in homomeric channels was −13.8 ± 3.2 mV in p.Pro285Thr, −12.2 ± 1.9 mV in p.Thr287Ile, −10.9 ± 1.8 mV in p.Gly281Glu, and −16.9 ± 2.0 mV in the *KCNQ2* WT. There was a right shift in the curve of V_1/2_ at 3.1 mV in p.Pro285Thr, at 4.7 mV in p.Thr287Ile [*t*(30) = −4.51, *p* < 0.001], and at 6 mV in p.Gly281Glu [*t*(28) = −5.84, *p <* 0.001], compared to that of the V_1/2_ in *KCNQ2* WT. The slope (K) was lower in p.Thr287Ile [8.3 ± 1.3, (mV/e-fold); *t*(30) = 2.59, *p =* 0.015] and p.Gly281Glu [8.0 ± 1.2, (mV/e-fold); *t*(28) = 2.27,conditiona 0.031] compared with that in *KCNQ2* WT [9.5 ± 2.2, (mV/e-fold)] (Table 2). The variant-to-WT currents ratio was 83.1% in homomeric p.Thr287Ile and 77.2% in homomeric p.Gly281Glu. In heteromeric-transfected *KCNQ2* WT cells and variants, the conductance–current curves were lowest in *KCNQ2* WT + p.Gly281 (Table 2). All three mutations in homomeric currents were lower in V_1/2,_ the K slope and currents (Table 2). After the transfection with the heteromeric *KCNQ2* WT and variants and with the *KCNQ3* WT (0.5 μg:0.5 μg:1 μg), the current amplitudes remained lower in the mutations (Table 2 and Appendix A).

In the heteromeric channels added with *KCNQ3* and *KCNQ2* WT, the p.Gly281Glu has lower normalized currents than the p.Thr287Ile and p.Pro285Thr at a −30 mV stimulation (Figure 5 and Appendix A).

Tail currents in the WT and mutations are shown (Figure 6A–C). The currents (pA) at a +40 mV potential showed that currents were lower in homomeric p.Gly281Glu (513.2 ± 64.7; *p* = 0.022) and in p.Thr287Ile (499.7 ± 34.6; *p* = 0.007) than the currents in KCNQ2 WT (625.6 ± 58.1) (Figure 6D). However, the tail currents of the p.Gly281Glu and p.Thr287Ile cells were increased in the KCNQ2 WT + p.Gly281Glu (656.5 ± 49.4) and in the KCNQ2 WT + p.Thr287Ile (646.6. ±87.65) at a +40 mV conditional voltage. In the heteromeric Kv7.2 + Kv7.3 and Kv7.2 + Kv7.3 + mutants, the currents were increased in p.Gly281Glu (1078.5 ± 153.8; *p* = 0.010) and in p.Thr287Ile (1111.6 ± 169.6; *p* = 0.033), and were still lower than in those in the KCNQ2 WT + KCNQ3 WT cells (1286.7. ±112.0) (Figure 6D).

### 4.5. Phenotypes, KCNQ2 Protein Expression, and Configuration Change on Cell Membranes

After analyzing *KCNQ2* protein expression for various variants, KCNQ2 protein expression on cell membranes did not differ significantly (*n* = 3) in *KCNQ2* WT, p.T287I and p.Gly281Glu (Figure 7A,B).

All p.Thr287Ile, p.Pro285Thr and p.Gly281Glu mutations are located in the SF domain of the *KCNQ2* channel. The computational model for p.Thr287Ile, p.Pro285Thr and p.Gly281Glu mutations predicted to change the configuration of the pore. For all three mutations, it was predicted that the diameter of the pores are different compared to those of the *KCNQ2* WT (Figure 8A,B).

### 4.6. Neurodevelopmental Outcomes Related to Mutations in the SF of KCNQ2

For the 35 mutations in the SF domain, using changes in amino acid weight between the WT and the *KCNQ2* mutations to predict EE resulted in 80.0% sensitivity, 80% specificity, a positive prediction rate of 96.0%, and a negative prediction rate of 40.0% (*p* = 0.006, χ^2^ (1, *n* = 35) = 7.56; odds ratio 16.0, 95% confidence interval, 1.50 to 170.63) (Table 3).

### 4.7. Neurodevelopmental Outcomes Related to Mutations in the SF of KCNQ2

Among the 35 mutations in the SF (Table 3 and Figure 5), the 5 that caused BFNCs were p.Asn258Ser, p.Asp259Thr, p.Asp259Tyr, p.Gly271Val and p.Tyr284Cys. Of those, 3 (60%; p.Asn258Ser, p.Asp259Thr, and p.Tyr284Cys) exhibited a mutated amino acid that was lighter than that of the WT (Table 3). The p.Gly271Val and p.Asp259Tyr mutations exhibited an increased molecular weight of the new amino acid. An analysis of pore diameters indicated relatively larger pores than in the others in the 5 mutations. Mutations that caused neonatal-onset EE represented the majority of mutations (85.7%), and they exhibited larger mutated protein weights (Table 3) and a smaller pore diameter than those in mutations that caused BFNCs (Table 3). This finding indicates that a high mutated amino acid weight could be an obstacle to pore size, a phenomenon that may be critical for determining neurodevelopmental outcomes. For the 35 mutations in the SF domain, using changes in amino acid weight between the WT and the *KCNQ2* mutations to predict EE resulted in 80.0% sensitivity, 80% specificity, a positive prediction rate of 96.0%, and a negative prediction rate of 40.0% (*p* = 0.006, χ^2^ (1, *n* = 35) = 7.56; odds ratio 16.0, 95% confidence interval, 1.50 to 170.63).

## 5. Discussion

The present study confirmed that *KCNQ2* and *KCNQ3* channels of p.Thr287Ile, p.Pro285Thr and p.Gly281Glu have dysfunctional effects on the Kv7.2 channel. Functional current changes were more severe in homomerically transfected p.Thr287Ile, p.Pro285Thr and p.Gly281. When concurrently heteromerically transfected with *KCNQ3* and *KCNQ2* mutants, the current changes were less severe but still lower in p.Thr287Ile, p.Pro285Thr and p.Gly281Glu than that of the WT. The pore loop between S5 and S6 contains a highly conserved SF that controls K+ permeability and selectivity [33]. Threonine (Thr) was hydrophilic and changed to isoleucine (Ile), which is hydrophobic. The p.Thr287Ile is located in the SF of the pore domain and can cause the *KCNQ2* protein pore domain to change according to the molecular model. The mutation (p.Thr287Ile) by the study matched the American College of Medical Genetics and Genomics (ACMG) criteria of PS2, PM1, PM2, PM5, PP4, PP3, and PS3. The p.Thr287Ile can be classified as pathogenic or likely pathogenic from uncertain significance. Of the three mutations (p.Thr287Ile, p.Gly281Glu and p.Pro285Thr), the conductance curves were similar, however, the p.Gly281Glu had worse *conductance* characters than the p.Thr287Ile and p.Pro285Thr when heteromerically transfected with *KCNQ3 + KCNQ2* + mutations. That is, mimicking the genetic balance in human. The finding was also compatible with the amino acid weight changes in p.Gly281Glu. The p.Gly281Glu has more amino acid weight changes than p.Thr287Ile and p.Pro285Thr (Table 3). The patient with p.Gly281Glu had worse clinical outcomes, including seizure frequencies and neurodevelopment, than patients with p.Thr287Ile and p.Pro285Thr. This finding increases our understanding of the association of *KCNQ2* EE with seizures, poor neurodevelopmental outcomes, ASD, and dyskinetic movement disorder beyond neonatal age despite seizure remission, and it could supplement related knowledge and improve the management of affected patients’ conditions.

The *KCNQ2* mutation phenotype of “severe or EE” missense variants were clustered at S4, S5, the pore loop that contains the SF, S6, prehelix A, helix B, and the helix B–C linker of Kv7.2 [33]. Mutations in the SF might affect the channel-gating function and contribute to severe phenotypes. In our case, when p.Gly281Glu (patient 2) and p.Gly281Arg [52] were compared, the outcome of patients with p.Gly281Glu was more favorable than that of those with p.Gly281Arg in terms of phenotype and the functional current results related to HEK293 cells. *KCNQ2* mutations affect the protein expression and M-current in the cells of the midbrain and striatum, and this is also a crucial factor in dyskinesia after the age of 4 weeks. In the patients presenting with EE, a transient signal change in the basal ganglia of the brain could be detected by an MRI in the neonatal period of approximately two-thirds of patients, but resolved at 2 to 4 years old [8]. More than 200 *KCNQ2* genotypes have been described thus far, but the phenotypes that persist after age 4 weeks are rarely reported. Neurodevelopmental outcomes such as cognition, language, life quality, and other reported behaviors should be further noted and managed by clinicians for the benefit of clinicians and parents.

Among the 35 mutations in the SF, the 5 that caused BFNCs were p.Asn258Ser, p.Asp259Thr, p.Asp259Tyr, p.Gly271Val and p.Tyr284Cys. Of those, 3 (60%; p.Asn258Ser, p.Asp259Thr, and p.Tyr284Cys) exhibited a mutated amino acid that was lighter than that of the WT (Table 3). The p.Gly271Val and p.Asp259Tyr mutations exhibited increased the molecular weight of the new amino acid. An analysis of the pore diameters indicated relatively larger pores than in the others in the 5 mutations. Mutations that caused neonatal-onset EE represented the majority of mutations (85.7%), and they exhibited larger mutated protein weights (Table 3) and a smaller pore diameter than those in mutations that caused BFNCs (Table 3). This finding indicates that a high mutated amino acid weight could be an obstacle to pore size, a phenomenon that may be critical for determining neurodevelopmental outcomes.

In neonatal-onset *KCNQ2* EE, an MRI can reveal transient basal ganglion abnormalities. *KCNQ2* exhibits immunoreactivity on the somata of dopaminergic and parvalbumin (PV)-positive (presumed GABAergic) cells of the substantia nigra, cholinergic large aspiny neurons of the striatum, and GABAergic and cholinergic neurons of the globus pallidus [15]. Thus, M-current dysfunction may contribute to the hyperactivity and network dysregulation characteristics of neonatal-onset EE, and *KCNQ2/3* channel regulation may be a target for therapeutic intervention [61]. However, reports of *KCNQ2*-associated movement disorder are rare. The selective openers of Kv7.2/3 channels might be candidates for the treatment of dyskinesias because antidyskinetic effects occurred at well-tolerated doses [62].

The most *KCNQ2* mutation are missense mutations. Truncated and splice-site mutations are the next most common mutations. The phenotypes and genotypes are complex. In general, nonsense, splice, and frameshifts cause a mild phenotype of familial *KCNQ2* BFNC. However, single mutations might manifest various clinical phenotypes within family members [63]. Changes in the functional current of the *KCNQ2* mutants were not necessarily correlated to the phenotype. As these mutations were localized on the P loop, the selectivity of the channel to K+ was reduced by variants, and channels became permeable to Na+. This could ultimately explain why global currents were slightly affected. The loss of function is the major mechanism in *KCNQ2* EE with de novo mutations. “Change of function” has been reported [64] recently in a *KCNT2* de novo mutation causing EE. There is also an alternative mechanism, particularly for mutations located in the SF region. Patients with de novo mutations and *KCNQ2* EE are associated with severe developmental delays. *KCNQ2* mutations found in the voltage sensor in S1–S4 or pore regions cause a more severe dominant–negative effect and lead to *KCNQ2* EE. Mutations located in the calmodulin domain of the C-terminal region were also reported to have more severe phenotypes. Some of the C-terminal mutations impair [65] surface expression by reducing protein stability or binding to calmodulin and thereby affecting transport to the surface membrane protein [28]. A study investigated a mutation (p.Lys526Asn in C-terminal), which caused the alteration of voltage-dependence of activation in these channels without changes in intracellular trafficking or plasma membrane expression [63]. The complex functions of the long C-terminal region exhibited interactions of syntaxin, phosphatidylinositol 4, 5-bisphosphate, ankyrin-G, Syn−1A, and the A-kinase anchoring protein [2,5,15,16,17,18,19,20,66]. However, the pore domain in the *KCNQ2* protein can affect channel gating and increase the threshold for channel activation without a significant channel number change [13,28,63,65]. The hypothesis is that mutations result in malfunctioning channels, and they do not affect the expression of cell KCNQ2 surface proteins.

This study has a few limitations. Due to the complexity of three dimensional graphics for the pore region, the predicted 3D model exhibited the pore region change by mutations. The determination of the phenotype is complex, and the phenotype may be due to the electrical charge of mutated proteins, the hydrophobic or hydrophilic characters of mutated proteins, modified genes, or acquired brain injury due to uncontrolled seizures. However, we found that the weight of mutated proteins can be a critical factor in mutations of the *KCNQ2* pore region. The predicted 3D structure denotes the effect of the mutation of the protein. Using HEK293 cells as a functional study in vitro with a potassium channel and the limitations of numbers of cells might contribute to the bias in currents, however, the numbers of cell are similar to other studies, which cause significant findings [52]. The nonexpression of potassium ion channels in HEK293 cells makes them an excellent model for whole-cell patch-clamp studies because only minor interfering currents occur. As a result, HEK293 cells have been widely used in cell biology, and the gene expression of HEK293 cells is similar to the gene expression of neurons.

## 6. Conclusions

These findings suggest that p.Thr287Ile, p.Pro285Thr. and p.Gly281Glu are pathogenic to *KCNQ2* EE and cause homomeric and heteromeric Kv7.2 current changes. All mutations cause neonatal EE and ASD beyond neonatal age. In the SF mutations of *KCNQ2*, patient outcomes are correlated with amino acid weight changes in the *KCNQ2* channel.

## Figures and Tables

**Figure 1 cells-11-00894-f001:**
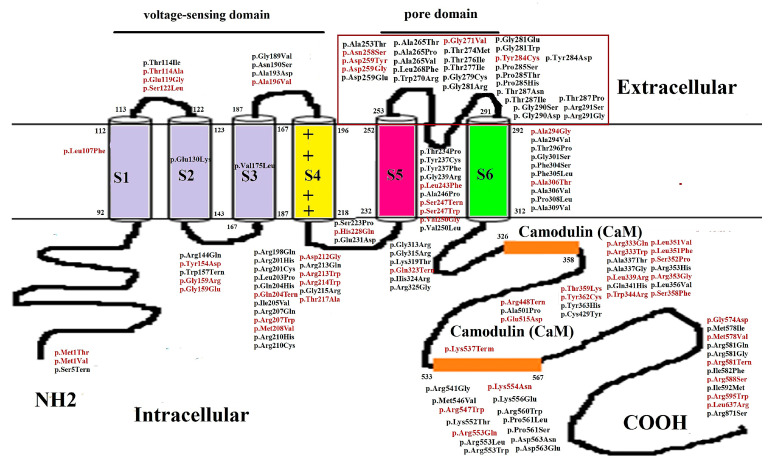
Mutations of *KCNQ2* are demonstrated in corresponding to KCNQ2 functional domains. Mutations with red fonts indicate the phenotype of BFNCs; mutations with black fonts are *KCNQ2* EE. Twenty-eight mutations (within the brown square) are in the SF domain of *KCNQ2*. Five mutations highlighted in red (17.9%) indicate those causing benign familial neonatal convulsions; 23 mutations highlighted in black (82.1%) indicate those causing epileptic encephalopathy.

**Figure 2 cells-11-00894-f002:**
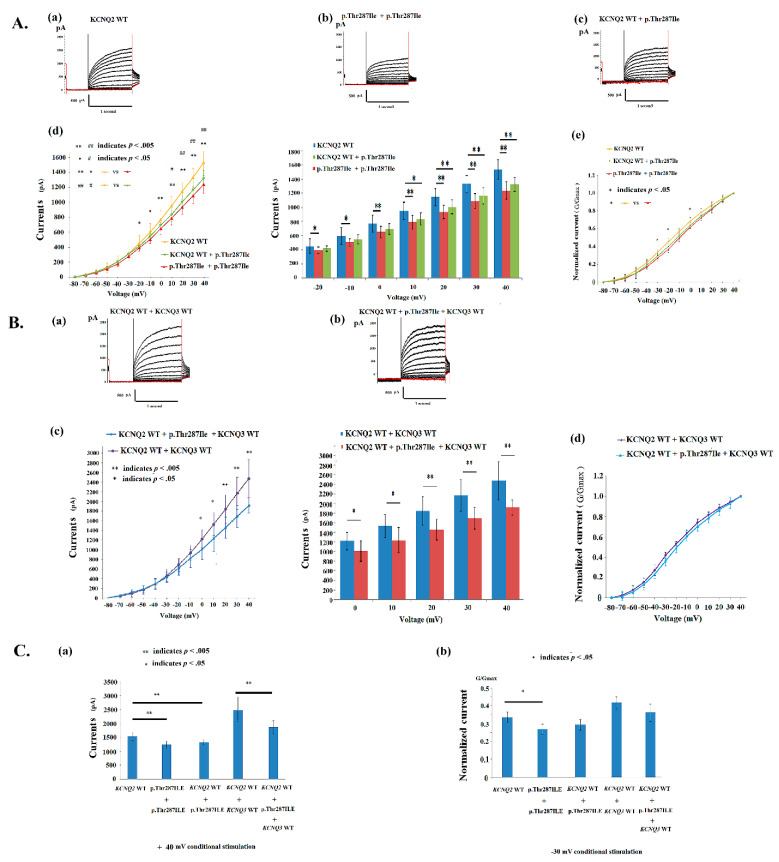
(**A**) Analysis of the electrophysiological properties of HEK293 cells in *KCNQ2* channels. The voltage-clamp steps were from −80 mV and +40 mV in 10-mV increments. The cells transfected with the (**a**) *KCNQ2* wild type (WT) (2 μg) (*n* = 22), (**b**) homomeric p.Thr287Ile (*n* = 10) (2 μg) and (**c**) heteromeric p.Thr287Ile + *KCNQ2* WT (1 μg:1 μg) (*n* = 10). (**d**) The conductance curves exhibited lower currents in homomeric p.Thr287Ile [*p* < 0.05 (*p* values see Appendix A); −20 to +40 mV] and heteromeric p.Thr287Ile + *KCNQ2* WT compared with *KCNQ2* WT [*p* < 0.05 (*p* values see Appendix A); 0 to +40 mV)]. * homomeric p.Thr287Ile versus *KCNQ2* WT; # p.Thr287Ile + *KCNQ2* WT versus *KCNQ2* WT. (**e**) The normalized currents in homomeric p.Thr287Ile were lower (G/G max) (*p* < 0.05; −30 to −20 mV and 0 mV) than those in cells with *KCNQ2* WT. (**B**) (**a**) The *KCNQ2* WT + *KCNQ3* WT (1 μg:1 μg) (*n* = 10) and (**b**) *KCNQ2* WT + p.Thr287Ile + *KCNQ3* WT (0.5 μg:0.5 μg:1 μg) (*n* = 10) were analyzed. (**c**) The conductance curves exhibited lower currents (*p* < 0.05 (*p* values in Appendix A); 0 to +40 mV) compared with *KCNQ2* WT + *KCNQ3* WT. (**d**) The normalized currents in p.Thr287Ile + *KCNQ2* WT + *KCNQ3* WT were lower (G/G max) than those in cells with *KCNQ2* WT + *KCNQ3* WT. (**C**) (**a**) At +40 mV conditional stimulation, the currents in *KCNQ2* WT and *KCNQ2* WT + *KCNQ3* WT are superior to homomeric p.Thr287Ile (*t* (30) = 5.66, *p* < 0.001), heteromeric p.Thr287Ile + *KCNQ2* WT (*t*(30) = 4.23, *p* < 0.001), and *KCNQ2* WT + p.Thr287Ile + *KCNQ3* WT (*t* (18) = 4.24, *p* < 0.001), correspondingly. (**b**) The normalized currents (G/Gmax) are different among channels, and significantly [*t* (30) = 2.39, *p =* 0.027; −30 mV] better in *KCNQ2* WT versus homomeric p.Thr287Ile. * and # indicate *p* < 0.05; ** and ## indicate *p* < 0.005.

**Figure 3 cells-11-00894-f003:**
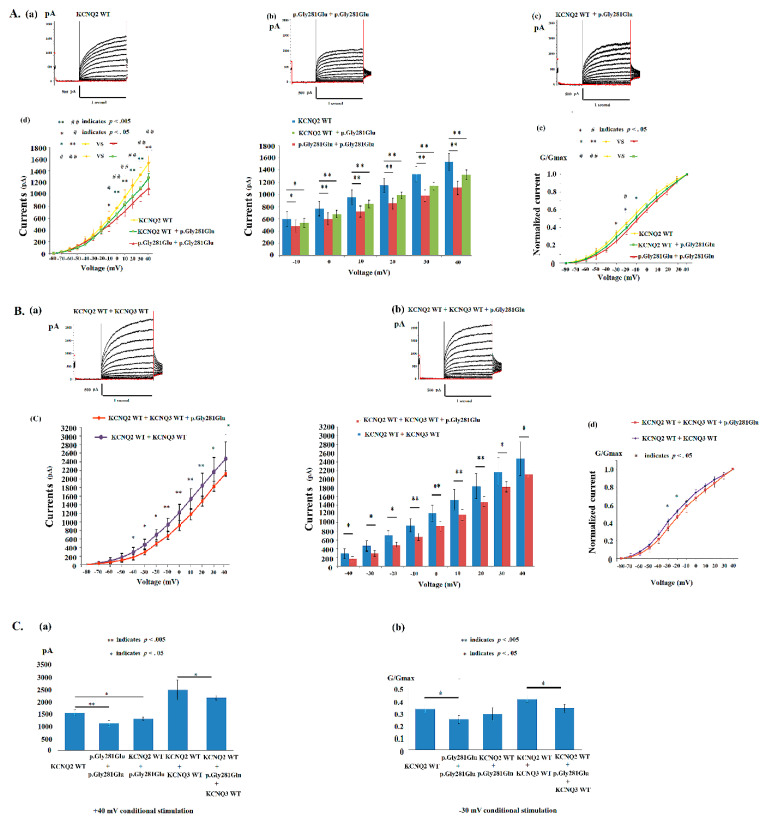
(**A**) The cells transfected with the (**a**) *KCNQ2* wild type (WT) (2 μg) (*n* = 22), (**b**) homomeric p.Gly281Glu (2 μg) (*n* = 8) and (**c**) heteromeric p.Gly281Glu + *KCNQ2* WT (1 μg:1 μg) (*n* = 8) were analyzed. The voltage-clamp steps were from −80 mV and +40 mV in 10-mV increments. (**d**) Cells transfected with p.Gly281Glu exhibited lower [*p* < 0.05 (*p* values see Appendix A); −10 + 40 mV) currents in homomeric p.Gly281Glu and in heteromeric p.Gly281Glu + *KCNQ2* WT [*p* < 0.05 (*p* values see Appendix A); −10 + 40 mV] compared with *KCNQ2* WT, correspondingly. * homomeric p. Gly281Glu versus *KCNQ2* WT; # p. Gly281Glu + *KCNQ2* WT versus *KCNQ2* WT. (**e**) The normalized currents in homomeric p. Gly281Glu were lower (G/G max) (*p* < 0.05; −30 to −10 mV) than those in cells with *KCNQ2* WT. (**B**) The (**a**) *KCNQ2* WT + *KCNQ3* WT (1 μg:1 μg) (*n* = 10) and (**b**) *KCNQ2* WT + p. Gly281Glu + *KCNQ3* WT (0.5 μg:0.5 μg:1 μg) (*n* = 8) were analyzed. (**c**) The *KCNQ2* WT + p.Gly281Glu + *KCNQ3* WT exhibited lower currents [*p* < 0.05 (*p* values see Appendix A)] compared with *KCNQ2* WT + *KCNQ3* WT from −40 to +40 mV conditional stimulation. (**d**) The normalized currents in p.Gly281Glu + *KCNQ2* WT + *KCNQ3* WT were lower (G/G max) than those in cells with *KCNQ2* WT + *KCNQ3* WT at −30 [t(16) = 2.81, *p* = 0.012] and −20 mV [t(16) = 2.72, *p* = 0.018] stimulation. (**C**) (**a**) At +40 mV stimulation, the currents in *KCNQ2* WT and *KCNQ2* WT + *KCNQ3* WT are superior to homomeric p. Gly281Glu (t(30) = 8.76, *p* < 0.001), heteromeric p. Gly281Glu + *KCNQ2* WT ((t(30) = 4.18, *p* < 0.001), and *KCNQ2* WT + p. Gly281Glu + *KCNQ3* WT (t(18) = 2.49, *p* = 0.024), correspondingly. (**b**) The value of G/Gmax was significantly low at a conditional voltage of −30 mV in homomeric p. Gly281Glu [*t*(28) = 2.48, *p =* 0.024] and heteromeric *KCNQ2* WT + p. Gly281Glu + *KCNQ3* WT (*t*(16) = 2.81, *p =*.012) compared with WT, correspondingly. * and # indicate *p* < 0.05; ** and ## indicate *p* < 0.005.

**Figure 4 cells-11-00894-f004:**
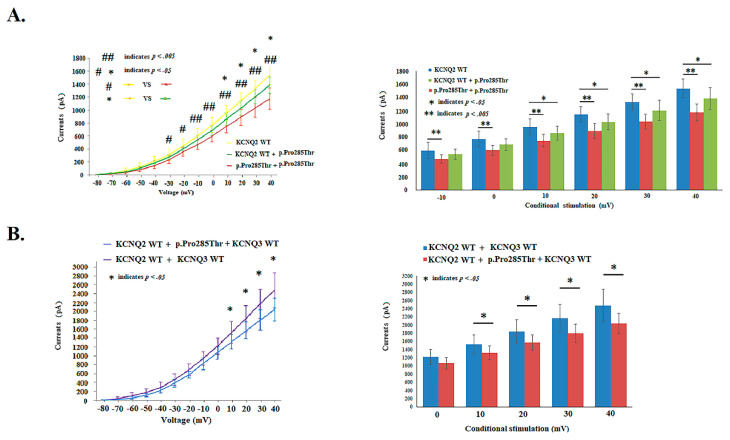
(**A**) Cells transfected with p.Pro285Thr exhibited lower [*p* < 0.05 (*p* values see Appendix A); −10 to +40 mV] currents in homomeric p.Pro285Thr and in heteromeric p.Pro285Thr + *KCNQ2* WT [*p* < 0.05 (*p* values see Appendix A); +10 to +40 mV] compared with *KCNQ2* WT, correspondingly. * and # indicate *p* < 0.05; ** and ## indicate *p* < 0.005. (**B**) The *KCNQ2* WT + p.Pro285Thr + *KCNQ3* WT exhibited lower current densities [*p* < 0.05 (*p* values see Appendix A)] compared with *KCNQ2* WT + *KCNQ3* WT from +10 to +40 mV conditional stimulation. * indicates *p* < 0.05.

**Figure 5 cells-11-00894-f005:**
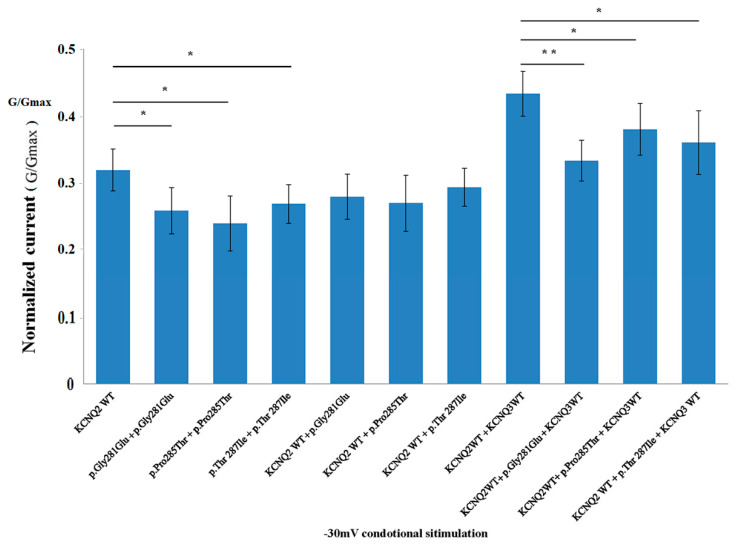
The normalized current in p.Thr287Ile, p.Gly281Glu, p.Pro285Thr and wild type demonstrated the homomerically transfections in 3 mutations that were significant lower than the wild type at −30 mV stimulations. In heteromerically transfected with *KCNQ3* + *KCNQ2* + variants, the p.Gly281Glu exhibited lower normalized currents than p.Thr287Ile, p.Pro285Thr and *KCNQ3 + KCNQ2* wild type. * indicates *p* < 0.05; **, *p* < 0.005. *KCNQ2* WT and mutant *KCNQ2* alleles were transfected into HEK293 cells in homomeric mutants (2 μg), heteromeric *KCNQ2* + mutants (1 μg + 1 μg), and heteromeric *KCNQ2* WT + variants + *KCNQ3* WT (0.5 μg:0.5 μg:1 μg), respectively.

**Figure 6 cells-11-00894-f006:**
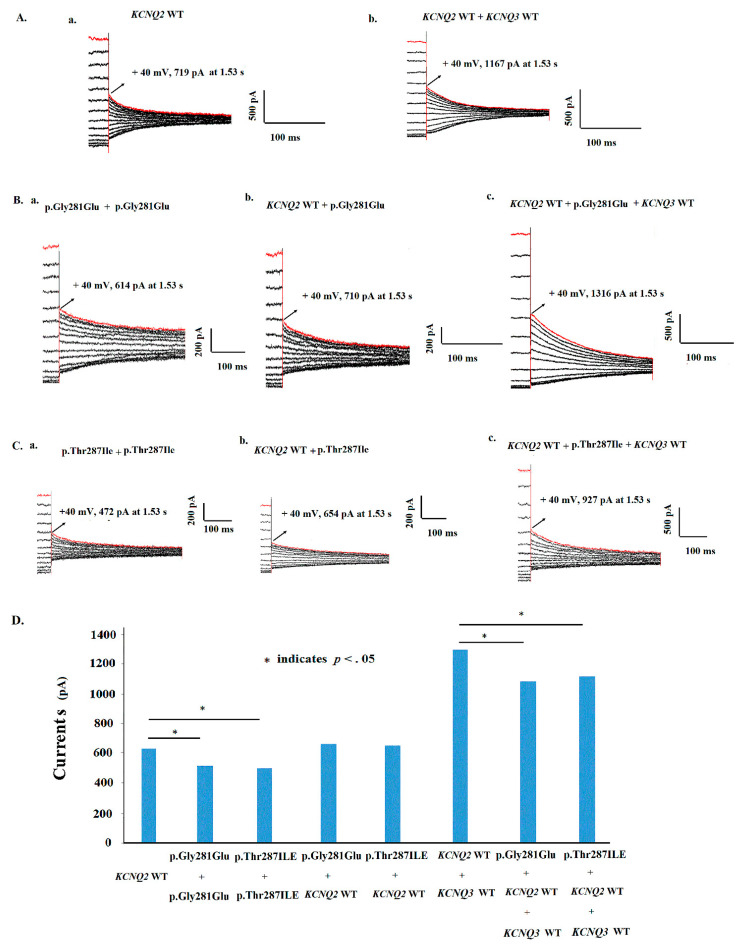
(**A**) Tail currents in the WT and mutations are shown at +40 mV conditional voltage. (**a**) KCNQ2 WT (2 μg) (*n* = 22), (**b**) KCNQ2 WT + KCNQ3 WT (1 μg:1 μg) (*n* = 10). (**B**) (**a**) The homomeric p.Gly281Glu (*n* = 8), (**b**) heteromeric p.Gly281Glu + KCNQ2 WT (1 μg:1 μg) (*n* = 8), (**c**) heteromeric KCNQ2 WT + p.Gly281Glu + KCNQ3 WT (0.5 μg:0.5 μg:1 μg) (*n* = 8). (**C**) (**a**) The homomeric p.Thr287Ile (2 μg) (*n* = 10), (**b**) heteromeric p.Thr287Ile + KCNQ2 WT (1 μg:1 μg) (*n* = 10), (**c**) heteromeric KCNQ2 WT + p.Thr287Ile + KCNQ3 WT (0.5 μg:0.5 μg:1 μg) (*n* = 10). (**D**) The homomeric variants had lower currents in p.Gly281Glu (513.2 ± 64.7; *p* = 0.022) and in p.Thr287Ile (499.7 ± 34.6; *p* = 0.007) than the currents in the KCNQ2 WT (625.6 ± 58.1). In the heteromeric Kv7.2 + Kv7.3 and Kv7.2 + Kv7.3 + mutants, the currents were lower in p.Gly281Glu (1078.5 ± 153.8; *p* = 0.010) and in p.Thr287Ile (1111.6 ± 169.6; *p* = 0.033), than in those in the KCNQ2 WT + KCNQ3 WT cells (1286.7 ± 112.0). * indicates *p* < 0.05.

**Figure 7 cells-11-00894-f007:**
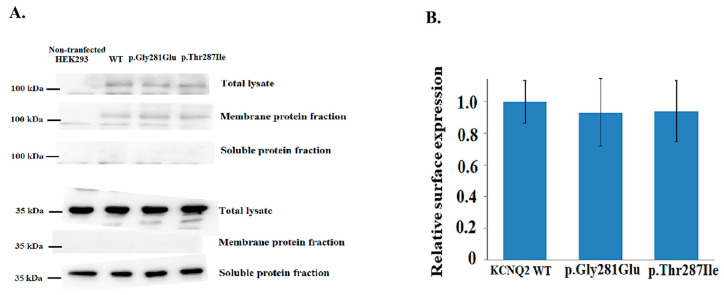
(**A**) Western blotting demonstrated the protein expression in *KCNQ2* WT, p.Thr287Ile, and p.Gly281Glu. *KCNQ2* protein expression was not significantly (*n* = 3) (Appendix A) different in p. T287I and p. Gly281Glu. Protein expression on cell membranes did not differ significantly for both mutations and *KCNQ2* WT. (**B**) Western blotting demonstrated the protein expression in *KCNQ2* WT, p.Thr287Ile, and p.Gly281Glu. KCNQ2 protein expression was not significant (*n* = 3).

**Figure 8 cells-11-00894-f008:**
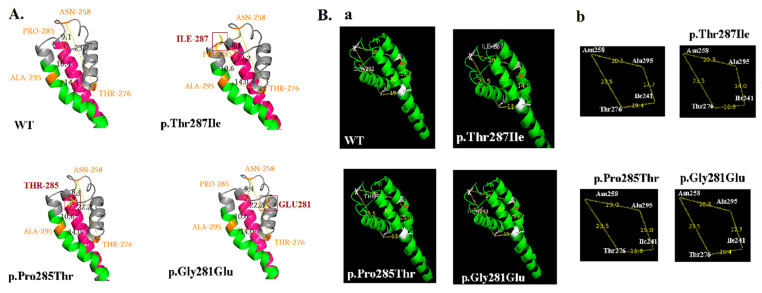
Schematic representation of the *KCNQ2* subunit with the position of the mutations of p.Thr287Ile. The predicted 3D model of the *KCNQ2* channel protein (c5vmsA_.1.pdb) was then used to analyze the structural differences between *KCNQ2* WT and mutations by using Swiss-PdbViewer and PyMOL, respectively. Mutation (p.Thr287Ile, p.Gly281Glu and p.Pro285Thr) sites at the selectivity filter (SF) might alter accessibility for potassium ions through the channels. The p.Thr287Ile, p.Gly281Glu and p.Pro285Thr are located in the SF of the pore domain and cause the *KCNQ2* protein configuration of pore domain change according to the molecular model. (**A**) The pore change was determined by calculating the distances from Asn (p.258) to Pro (p.285), Pro (p.285) to Ala (p.295), Ala (p.295) to Thr (p. 276), and Thr (p.276) to Asn (p.258), respectively. The diameters of the *KCNQ2* channel pores were determined by calculating the distance from protein 258 to 285 (A), 285 to 295 (B), 295 to 276 (C), and 276 to 258 (D). The pore configuration was expressed as A × B × C × D. The mutations change the configurations of pores in p.Thr287Ile (from 31,617.3 in WT changed to 27,673.6), in p.Gly281Glu (from 31,617.3 changed to 27,673.6) and p.Pro285Thr (from 31,617.3 to 27,673.6). (**B**) (**a**) The pore change were also determined by calculating the distances from Asn (p.258) to Thr (p.276), Thr (p.276) to Ile (p.241), Ile (p.241) to Ala (p.295), and Ala (p.295) to Asn (p.258), respectively. (**b**) The pore configurations were changed by the mutations of p.Thr287Ile and p.Pro285Thr. Mutations changed configurations of pores in p.Thr287Ile, p.Gly281Glu, and p.Pro285Thr. The diameter of the *KCNQ2* channel pore was determined by calculating the distance from protein 258 to 276 (A1), 276 to 295 (B1), 241 to 276 (C1), and 241 to 295 (D1). The pore distance was expressed as A1 × B1 × C1 × D1. Mutations changed the configurations of pores in p.Thr287Ile (from 163,809.4 in the WT to 78,808.7), in p.Gly281Glu (not different from 163,809.4 in WT), and p.Pro285Thr (from 163,809.4 in the WT to 77,691.0).

**Table 1 cells-11-00894-t001:** Clinical presentations and long-term neurodevelopmental outcomes in three *KCNQ2* mutations located in SF domain.

	Patient 1	Patient 2	Patient 3
**Genotype in patients**	p.Thr287Ile	p.Gly281Glu	p.Pro285Thr
**Inheritance**	*De novo*	*De novo*	*De novo*
**Functional domain**	Selectivity filter	Selectivity filter	Selectivity filter
**Family history**	No	No	No
**First seizure day**	Day 3	Day 3	Day 2
**Seizure frequency before drug control**	Daily	Daily	Daily
**Age when seizure-free**	Partial remission of seizures at 4 months, with recurrent febrile seizures	No remission of seizures	Partial remission of seizures after 1 year
**MRI**	Unremarkable at 4 years old	Thin corpus callosum, brain atrophy at 4 years old	Thin corpus callosum
**Long-term neurodevelopmental outcomes**	Lack of language production, can sit, inability to walk, severe cognitive disability at 5 years old.	Lack of language production, cannot sit without support, inability to walk, severe cognitive disability at 5 years old.	Lack of language, can sit, inability to walk, severe cognitive disability at 4 years old.

The sequence data of each patient were checked against the GenBank reference sequence and version number of *KCNQ2* gene (NM_172107.3). MRI, magnetic resonance imaging.

**Table 2 cells-11-00894-t002:** Electrophysiological study in three *KCNQ2* mutations in SF domain.

HEK293 Transfection	KCNQ2 WT	*KCNQ2* WT + *KCNQ3* WT	p.Gly281Glu	p.Thr287Ile	p.Pro285Thr
**Homomeric variants (2 μg)**					
V_1/2_ (mV) (mean ± SD) (N) (*p*)	−16.9 ± 2.0 (22)		**−10.9 ± 1.8 ** (8)** **(*p* < 0.001)**	**−12.2 ± 1.9 ** (10)** **(*p* < 0.001)**	**−13.8 ± 3.2 * (10)** **(*p* = 0.012)**
K (mV/e-fold) (mean ± SD) (N) (*p*)	9.5 ± 2.2 (22)		**8.0 ± 1.2 * (8)** **(*p* = 0.031)**	**8.3 ± 1.3 * (10)** **(*p* = 0.015)**	**7.9 ± 0.9 ** (10)** **(*p* = 0.009)**
Currents (pA) (mean ± SD) (N) (*p*)	579.8 ± 46.0 (22)		**447.7 ± 63.7 ** (8)** **(*p* < 0.001)**	**481.8 ± 56.9 * (10)** **(*p* = 0.015)**	**436.0 ± 55.0 ** (10)** **(*p* <. 001)**
**Heteromeric variants** **(*KCNQ2* + mutants) (1 μg:1 μg)**					
V_1/2_ ^(mV)^ (mean ± SD) (N) (*p*)			**−13.0 ± 2.1 * (8)** **(*p* = 0.010)**	**−13.9 ± 1.5 * (10)** **(*p* = 0.011)**	−14.8 ± 2.1 (10) (*p* = 0.073)
K (mV/e-fold) (mean ± SD) (N) (*p*)			8.5 ± 1.6 (8) (*p* = 0.159)	8.7 ± 1.1 (10) (*p* = 0.133)	8.8 ± 1.5 (10) (*p* = 0.351)
Currents (pA) (mean ± SD) (N) (*p*)			**490.0 ± 38.0 * (8)** **(*p* = 0.016)**	519.4 ± 52.0 (10) (*p* = 0.056)	518.0 ± 24.5 (10) (*p* = 0.74)
**Heteromeric variants** **(*KCNQ2* + mutants + *KCNQ3*)** **(0.5 μg:0.5 μg:1μg)**					
V_1/2_ ^(mV)^ (mean ± SD) (N) (*p*)		−20.8 ± 1.6 (10)	−18.5 ± 2.0 (8) (*p* = 0.054)	−19.3 ± 1.4 (10) (*p* = 0.122)	−18.8 ± 2.2 (10) (*p* = 0.061)
K (mV/e-fold) (mean ± SD) (N) (*p*)		14.9 ± 3.0 (10)	**11.1 ± 0.7 ** (8)** **(*p* = 0.002)**	12.5 ± 2.7 (10) (*p* = 0.081)	13.1 ± 1.7 (10) (*p* = 0.107)
Currents (pA) (mean ± SD) (N) (*p*)		813.9 ± 118.3 (10)	705.0 ± 64.4 (8) (*p* = 0.051)	753.3 ± 94.2 (10) (*p* = 0.053)	715.2 ± 58.3 (10) (*p* = 0.054)

WT, wild type; V is the test potential; V½, half-maximal activation voltage; SD, standard deviation. The data were then fitted to a Boltzmann distribution of the following form: *G**/Gmax* = 1/(1 + *exp*[(*V − V*½)/*dx*]). Current was expressed as average currents (pA) from −80 to +40 mV in the transfected cells. * indicates *p* < 0.05 compared with *KCNQ2* WT; **, *p* < 0.005 compared with *KCNQ2* WT. Bold font indicates significantly different from WT. Data rounded off to the first decimal place. Homomeric transfected variants and heteromeric transfected *KCNQ2* WT + variants were compared with the current in *KCNQ2* WT (2 μg), respectively. Heteromeric *KCNQ2* WT + *KCNQ3* WT + variants were compared with the current in *KCNQ2* WT + *KCNQ3* WT (1μg:1 μg), respectively.

**Table 3 cells-11-00894-t003:** Among 35 mutations in the selectivity filter domain, changes in amino acid weight between the WT and the mutations are demonstrated.

KCNQ2 Genotype ^#^	WT	Weight of Amino Acid of KCNQ2 WT	Amino Acid Polarity	Amino Acid Change	Weight of Mutated Amino Acid	Mutated Amino Acid Polarity	Phenotype	References
Ala253Thr	Ala	89	N	Thr	119	Y	EE	[44]
Gly256Trp	Gly	75	N	Trp	204	Y	EE	[29]
**Asn258Ser**	**Asn**	**132**	**Y**	**Ser**	**105**	**Y**	**BFNC**	[45]
Asp259Tyr	Asp	133	Acid R group	Tyr	181	Y	BFNC	[46]
**Asp259Gly**	**Asp**	**133**	**Acid R group**	**Gly**	**75**	**N**	**BFNC**	[47]
Asp259Glu	Asp	133	Acid R group	Glu	147	Acid R group	EE	[29]
Ala265Val	Ala	89	N	Val	117	N	EE	[48]
Ala265Pro	Ala	89	N	Pro	115	N	EE	[7]
Ala265Thr	Ala	89	N	Thr	119	Y	EE	[49]
Leu268Phe	Leu	131	N	Phe	165	N	EE	[50]
**Trp269Leu**	**Trp**	**204**	**Y**	**Leu**	**131**	**N**	**EE**	[51]
**Trp270Arg**	**Trp**	**204**	**Y**	**Arg**	**174**	**Basic R group**	**EE**	[52]
Gly271Val	Gly	75	N	Val	117	N	BFIC	[53]
Ile273Asn	Ile	131	N	Asn	132	Y	EE	[51]
Thr274Met	Thr	119	Y	Met	149	N	EE	[7]
Thr276Ile	Thr	119	Y	Ile	131	N	EE	[54]
Thr277Ile	Thr	119	Y	Ile	131	N	EE	[55]
Gly279Cys	Gly	75	N	Cys	121	Y	EE	[44]
Gly281 Trp	Gly	75	N	Trp	204	Y	EE	[50]
Gly281Arg	Gly	75	N	Arg	174	Basic R group	EE	[8]
Gly281Glu	Gly	75	N	Glu	147	Acid R group	EE	The study
Gly281Trp	Gly	75	N	Trp	204	Y	EE	[50]
Asp282Asn	Asp	133	Acid R group	Asn	132	Y	EE	**[56]**
**Tyr284Asp**	**Tyr**	**181**	**Y**	**Asp**	**133**	**Acid R group**	**EE**	[46]
**Tyr284Cys**	**Tyr**	**181**	**Y**	**Cys**	**121**	**Y**	**BFNC**	**[3]**
Pro285Thr	Pro	115	N	Thr	119	Y	EE	The study
**Pro285Ser**	**Pro**	**115**	**N**	**Ser**	**105**	**Y**	**EE**	**[57]**
Pro285His	Pro	115	N	His	155	Basic R group	EE	[6]
Thr287Ile	Thr	119	Y	Ile	131	N	EE	The study
**Thr287Pro**	**Thr**	**119**	**Y**	**Pro**	**115**	**N**	**EE**	**[58]**
Thr287Asn	Thr	119	Y	Asn	132	Y	EE	[49]
Gly290Asp	Gly	75	N	Asp	133	Acid R group	EE	[7]
Gly290Ser	Gly	75	N	Ser	105	Y	EE	[49]
**Arg291Ser**	**Arg**	**174**	**Basic R group**	**Ser**	**105**	**Y**	**EE**	**[59]**
**Arg291Gly**	**Arg**	**174**	**Basic R group**	**Gly**	**75**	**N**	**EE**	**[60]**

WT indicates wild type; N, non-polar amino acids; Y, polar amino acids; BFNC, benign familial neonatal convulsions; EE, epileptic encephalopathy. Bold indicates the mutated amino acid that was lighter than that of the WT. **^#^** Among 35 mutations in the SF domain, using changes in amino acid weight between the WT and the *KCNQ2* channel to predict EE resulted in 80.0% sensitivity, 80% specificity, a positive prediction rate of 96.0%, and a negative prediction rate of 40.0% (*p* = 0.006, χ^2^ (1, *n* = 35) = 7.56; odds ratio 16.0, 95% confidence interval, 1.50 to 170.63).

## Data Availability

The datasets used and/or analyzed during the current study are available from the corresponding author on reasonable request.

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
