# Peer review of "KCNQ2 Selectivity Filter Mutations Cause Kv7.2 M-Current Dysfunction and Configuration Changes Manifesting as Epileptic Encephalopathies and Autistic Spectrum Disorders"

_cells, 2022, doi:10.3390/cells11050894_

Round 1

Reviewer 1 Report

General comment

Recent progress in modeling of basic structures of KCNQ channels is associated with rising attention for the role of alterations in the amino acid  sequence of the channel proteins and its functional consequences in clinical pathologies.

 The authors investigated, especially, the role of KCNQ channels in neonatal brain development. During a clinical screening, three patients with de novo mutations in the selective filter-domain of the KCNQ2 subtype of the channel were identified. This subtype is presumed to be threatened, especially, by haploinsufficiency with associated phenotypic functional epileptiform disturbances. The clinical time course reflected among the three patients with  de novo mutations only in one patient a remission of the neonatal seizure in response to standard drug regimens, but for all three cases long term cognitive impairment, delay  in language and motoric development observed up to the age of 3 to 5 years.

The authors expressed KCNQ2 wild types and homomeric and heteromeric variants of the mutants in HEK 293 cells and subjected these models to patch clamp investigations and immunohistological characterization of the channels.

Electrophysiological investigation revealed functional impairment of the mutant subtypes compared with the wild types of KCNQ2 and KCNQ3 subtypes. At the end the paper,the authors presented the results of a literature study on KCNQ2 channel mutations and modeling data reflecting potential consequences of the investigated mutations for the accessibility of the channels for potassium ions.

All together, the authors provide with their clinical and experimental investigations an interesting contribution for better understanding of the role of KCNQ2 channels in the occurrence of neonatal epileptiform disturbances with potential for improved approaches to their treatment.

Major comments

I wonder why the authors did not choose the other direction for presentation of their data, starting from their theoretical approach and knowledge on basic structure of KCNQs and general presence of mutations in KCNQ , the positioning of the mutations in the selective filter and the clinical consequences. For instance, Figure 6 would belong rather to the Introduction than to the Results because the data came from other authors ( see also Table 3). That would supply also for the readers which are less familiar with the channel topics the basic knowledge necessary for understanding of the approach. Then the  special population of neonates and children investigated here could  have been shown  followed by the  presentation of  the data and results of investigations about  the three de novo mutations, demonstration of the expression  of the different channel variants at protein level with  immune histological methods and characterization of homomeric and heteromeric variants with electrophysiological methods. Table 3 would better suit to the discussion than to the results. Finally , also the modeling data could have been shown.

Please add the respective numbers to the part of the text  as commonly used in this journal.

Please, use the recommendations given in the respective template or authors guidelines of the journal.

Please, separate also in Methods and Materials  the clinical part, electrophysiological and immunohistological part below their own headlines.

Minor comments

Page 2, Paragraph 2, line 13: (Ambrosio et al….)  Please add year of occurrence.

Page 3 Methods and Materials , chapter: Mutations of KCNQ2….

Here the authors refer to  their review of elaborative literature. Is this related to literature in the Introduction or in the paper in general or can there be provided a PRISMA-overview on the general approach to literature selection for this paper.

Page 5, Ethics, last line: ..obtained from parents of all patients.

Page 8 : line 4:   typos   : shifting instead of sifting

Please, check for similar typos.

Page 9 to 12 . The electrophysiological data for the three mutants are very extensively presented with figures, respective legends and again in the text. Perhaps it is possible to shorten the text and focus there more on the facts which the authors which to highlight.

Page 12, Figure 3: It could be helpful if the authors would give a schematic add on of the different homomeric and heteromeric approaches employed in de electrophysiological experiments.

Page 17, Figure 6, legend: It is not clear if the authors discovered all these mutations or if they are  known from literature. In case that the source are, for instance, reference in table 3 the authors should add .for source of data see table 3.

(also in the Abstract it is not clear which of the mutations have been discovered by the authors and which are literature data.)

Page 18 , line 8 :  and line 13: the author report on transient  MRI signal change in the basal ganglia  of the brains they found described in a report by Weckhuysen et al. 2013. Which of transient alteration was that and what was its duration?

Please, add a list of abbreviations.

Author Response

Feb. 10 (cells-1445147)

Reviewer 1

General comment

  • Recent progress in modeling of basic structures of KCNQ channels is associated with rising attention for the role of alterations in the amino acid  sequence of the channel proteins and its functional consequences in clinical pathologies.

 The authors investigated, especially, the role of KCNQ channels in neonatal brain development. During a clinical screening, three patients with de novo mutations in the selective filter-domain of the KCNQ2 subtype of the channel were identified. This subtype is presumed to be threatened, especially, by haploinsufficiency with associated phenotypic functional epileptiform disturbances. The clinical time course reflected among the three patients with  de novo mutations only in one patient a remission of the neonatal seizure in response to standard drug regimens, but for all three cases long term cognitive impairment, delay  in language and motoric development observed up to the age of 3 to 5 years.

The authors expressed KCNQ2 wild types and homomeric and heteromeric variants of the mutants in HEK 293 cells and subjected these models to patch clamp investigations and immunohistological characterization of the channels.

Electrophysiological investigation revealed functional impairment of the mutant subtypes compared with the wild types of KCNQ2 and KCNQ3 subtypes. At the end the paper,the authors presented the results of a literature study on KCNQ2 channel mutations and modeling data reflecting potential consequences of the investigated mutations for the accessibility of the channels for potassium ions.

All together, the authors provide with their clinical and experimental investigations an interesting contribution for better understanding of the role of KCNQ2 channels in the occurrence of neonatal epileptiform disturbances with potential for improved approaches to their treatment.

Reply: We are grateful for the opportunity to improve our manuscript and we thank the editorial board and the reviewers for their thoughtful and helpful comments and criticisms. We have modified the paper as suggested. Following are our point-by-point responses. We have also highlighted the principal changes in the “Revised Text”.

Major comments

  • I wonder why the authors did not choose the other direction for presentation of their data, starting from their theoretical approach and knowledge on basic structure of KCNQs and general presence of mutations in KCNQ , the positioning of the mutations in the selective filter and the clinical consequences. For instance, Figure 6 would belong rather to the Introduction than to the Results because the data came from other authors ( see also Table 3). That would supply also for the readers which are less familiar with the channel topics the basic knowledge necessary for understanding of the approach. Then the special population of neonates and children investigated here could  have been shown  followed by the  presentation of  the data and results of investigations about  the three de novo mutations, demonstration of the expression  of the different channel variants at protein level with  immune histological methods and characterization of homomeric and heteromeric variants with electrophysiological methods. Table 3 would better suit to the discussion than to the results. Finally , also the modeling data could have been shown.

Reply: Thanks for the opinion. 1. We have moved the Figure 6 to Introduction ( Figure 1), and changed to the order of figures. 2. We keep the Table 3 in the Result because that we have analyzed the character of the amino acid in the mutations that demostrated the result in the Result section (Neurodevelopmental Outcomes Related to Mutations in the SF of KCNQ2). 3. We moved some sentences in Results ( Neurodevelopmental Outcomes Related to Mutations in the SF of KCNQ2) to the section of Discussion.

In Introduction, paragraph 5, we added the sentences

The SF in KCNQ2 is from protein 253 to 291 (Figure 1). We found 35 mutations [5 caused BFNCs (14.3%) and 30 caused EE (85.7%)] according to a comprehensive literature review of literatures. All neurodevelopmental outcomes related to BFNCs were favorable. Among patients with mutations causing EE, the outcomes were poorer than the outcomes of those with mutations causing BFNCs and varied from moderate or severe to profound developmental disabilities and early mortality.”

In Result,

Neurodevelopmental Outcomes Related to Mutations in the SF of KCNQ2, changed to:

                   Neurodevelopmental Outcomes Related to Mutations in the SF of KCNQ2

“ For the 35 mutations in the SF domain, using changes in amino acid weight between the WT and the KCNQ2 mutations to predict EE resulted in 80.0% sensitivity, 80% specificity; a positive prediction rate of 96.0%, and a negative prediction rate of 40.0% (p = 0.006, χ2 (1,N=35) = 7.56; odds ratio 16.0, 95 % confidence interval, 1.50 to 170.63).”

In Discussion, paragraph 3, we added:

                             Among the 35 mutations in the SF, the 5 that caused BFNCs were p.Asn258Ser, p.Asp259Thr, p.Asp259Tyr, p.Gly271Val and p.Tyr284Cys. Of those, 3 (60%; p.Asn258Ser, p.Asp259Thr, and p.Tyr284Cys) exhibited mutated amino acid that was lighter than that of the WT (Table 3). The p.Gly271Val and p.Asp259Tyr mutations exhibited increased molecular weight of the new amino acid. An analysis of pore diameter indicated a relatively larger pores than in the others in the 5 mutations. Mutations that caused neonatal-onset EE represented the majority of mutations (85.7%), and they exhibited larger mutated protein weights (Table 3) and a smaller pore diameter than those in mutations that caused BFNCs (Table 3). This finding indicates that high mutated amino acid weight could be an obstacle to pore size, a phenomenon that may be critical for determining neurodevelopmental outcomes.

  • Please add the respective numbers to the part of the text as commonly used in this journal.
  •  

Reply: The respective numbers were added. We have revised the text by the template supported by the Journal.

  • Please, use the recommendations given in the respective template or authors guidelines of the journal.
  •  

Reply: We have revised the text that was supported by the Editor Office in the submission system.

  • Please, separate also in Methods and Materials  the clinical part, electrophysiological and immunohistological part below their own headlines.

Reply: We have changed it accordingly.

Minor comments

  • Page 2, Paragraph 2, line 13: (Ambrosio et al….)  Please add year of occurrence.

Reply: We have corrected it.

  • Page 3 Methods and Materials , chapter: Mutations of KCNQ2….

Here the authors refer to their review of elaborative literature. Is this related to literature in the Introduction or in the paper in general or can there be provided a PRISMA-overview on the general approach to literature selection for this paper.

Reply:  We reviewed the database in Human Gene Mutation Database (HGMD) (http://www.hgmd.cf.ac.uk/ac/index.php) and National Center for Biotechnology Information (NCBI) and confirmed the pathogenic characters of the mutations to review the corresponding literatures. We collected the missense mutations of KCNQ2. The phenotypes were classified to BFNC and KCNQ2 EE.

In Method,

Mutations of KCNQ2 in corresponding to KCNQ2 functional domains and phenotypes,

Changed to:

Mutations of KCNQ2 in corresponding to KCNQ2 functional domains and phenotypes

We reviewed the database in HGMD and NCBI and confirmed the pathogenic characters of the mutations to review the corresponding literatures. We collected the missense mutations of KCNQ2. The phenotypes were classified to BFNC and KCNQ2 EE. 

  • Page 5, Ethics, last line: ..obtained from parents of all patients.

Reply: We have corrected it.

  • Page 8 : line 4:   typos   : shifting instead of sifting

Reply: We have corrected it.

  • Please, check for similar typos.

Reply: We have corrected it and used the proofreading and editing services of professional native English speaking technical editors for both the first submission and this revision.

  • Page 9 to 12 . The electrophysiological data for the three mutants are very extensively presented with figures, respective legends and again in the text. Perhaps it is possible to shorten the textt and focus there more on the facts which the authors which to highlight.

Reply: We have shortened the Result.

  • Page 12, Figure 3: It could be helpful if the authors would give a schematic add on of the different homomeric and heteromeric approaches employed in de electrophysiological experiments.
  •  

Reply: Figure 5 (old Figure 3) legend have been changed to:

Figure 5. The normalized current in p.Thr287Ile, p.Gly281Glu, p.Pro285Thr and wild type demonstrated the homomerically transfections in 3 mutations were significant lower than the wild type at -30 mV stimulations. In heteromerically transfected with KCNQ3 + KCNQ2 + variants, the p.Gly281Glu exhibited lower normalized currents than p.Thr287Ile, p.Pro285Thr and KCNQ3 + KCNQ2 wild type. *indicates p < 0.05; **, p < 0.005. KCNQ2 WT and mutant KCNQ2 alleles were transfected into HEK293 cells in homomeric mutants (2 ug), heteromeric KCNQ2 + mutants (1 ug + 1 ug), and heteromeric KCNQ2 WT + variants + KCNQ3 WT(0.5 μg : 0.5 μg : 1 μg) respectively..

  • Page 17, Figure 6, legend: It is not clear if the authors discovered all these mutations or if they are  known from literature. In case that the source are, for instance, reference in table 3 the authors should add for source of data see table 3.

Reply: We changed the Figure 6 to the new Figure 1 as the reviewer’s suggestion.

In Figure 1, legend, Changed to:

Figure 1. Mutations of KCNQ2 are demonstrated in corresponding to KCNQ2 functional domains. Mutations with red fonts indicate the phenotype of BFNCs; mutations with black fonts are KCNQ2 EE (for sources of data, see Table 3). Twenty-eight mutations (within the brown square) are in the SF domain of KCNQ2. Five mutations highlighted in red (17.9%) indicate those causing benign familial neonatal convulsions; 23 mutations highlighted in black (82.1%) indicate those causing epileptic encephalopathy.

  • (also in the Abstract it is not clear which of the mutations have been discovered by the authors and which are literature data.)

Reply: All three patients are in our data, not from literature.

In Abstract, line 3-7, changed to:

“Three patients with neonatal EE carry de novo heterozygous KCNQ2 p.Thr287Ile, p.Gly281Glu and p.Pro285Thr, and all are with follow-up in our clinic.”

  • Page 18 , line 8 :  and line 13: the author report on transient  MRI signal change in the basal ganglia  of the brains they found described in a report by Weckhuysen et al. 2013. Which of transient alteration was that and what was its duration?

Reply: The duration of transient MRI change is not consistent. In our case, 3 MRI did not detect the transient abnormies

Reply: Not all patients were detected the MRI change. In the patients presenting with EE, transient signal change in the basal ganglia of the brain could be detected by MRI in the neonatal period of approximately two-thirds of patients, but resolved in 2 to 4 years old (Kato et al., 2013; Weckhuysen et al., 2013).

In Discussion, paragraph 2, changed to:

The KCNQ2 mutation phenotype of “severe or EE” missense variants were clustered at S4, S5, the pore loop that contains the SF, S6, prehelix A, helix B, and the helix B–C linker of Kv7.2 (J. Zhang et al., 2020). Mutations in the SF might affect the channel-gating function and contribute to severe phenotypes. In our case, when p.Gly281Glu (patient 2) and p.Gly281Arg (Gomis-Perez et al., 2019) were compared, the outcome of patients with p.Gly281Glu was more favorable than that of those with p.Gly281Arg in terms of phenotype and the functional current results related to HEK293 cells. KCNQ2 mutations affect the protein expression and M-current in the cells of the midbrain and striatum, and this is also a crucial factor in dyskinesia after age 4 weeks. In the patients presenting with EE, transient signal change in the basal ganglia of the brain could be detected by MRI in the neonatal period of approximately two-thirds of patients, but resolved in 2 to 4 years old (Kato et al., 2013; Weckhuysen et al., 2013). More than 200 KCNQ2 genotypes have been described thus far, but the phenotypes that persist after age 4 weeks are rarely reported. Neurodevelopmental outcomes such as cognition, language, life quality, and other reported behaviors should be further noted and managed by clinicians for the benefit of clinicians and parents.

  • Please, add a list of abbreviations.

Reply: We have added List of abbreviations.

List of abbreviations.

ACMG; American College of Medical Genetics and Genomics; ASD, autistic spectrum disorder; BFNCs, benign familial neonatal convulsions; DMEM, Dulbecco’s modified Eagle’s medium; EE, EDTA, ethylenediaminetetraacetic acid; EE, epileptic encephalopathy; EEG, electroencephalogram; FBS, fetal bovine serum; HGMD, Human Gene Mutation Database; K, slope; MRI, magnetic resonance image; NCBI, National Center for Biotechnology Information; PMSF, phenylmethyl sulfonyl fluoride; PV, parvalbumin; PVDF, polyvinylidene difluoride; OR, odds ratio; OXC, oxcarbazepine; SF , selectivity filter; V1/2, half-activation potential; WT, wild type.

Reviewer 2 Report

In the manuscript untitled “KCNQ2 selectivity filter mutations cause Kv7.2 M current dysfunction and configuration changes manifesting as epileptic encephalopathies and autistic spectrum disorders” , Lee and Collaborators describe the clinical presentation  of three patients harboring the variants p.Thr287Ile, p.Gly281Glu, p.Pro285Thr of the Kv7.2 subunit and the functional consequence of these variants  on M current carried by homomeric Kv7.2 channels and by heteromeric Kv7.3 and Kv7.2 channels in configuration that mimic patient’s situation. In addition authors performed computational protein analysis of the mutations. The three patients harboring these mutations exhibited developmental delay including dyskinetic movement disorders and ASD . The goal of this study was to see if there is any phenotype-genotype correlation between the functional impact of the variant and the disease severity. Although such type analysis is not new,  this study describe the functional impact of three new variants in KCNQ2-EE.  I have some concerns regarding this study.

1) Important informations are missing in the electrophysiological study: Kinetics of activation and deactivation and reversal potential of the current are not shown. In addition the effect of the 3 mutations on current density is not so strong (< 20% of decrease). This decrease is similar to what is observed in BFNS. Therefore the amount of current decrease is not predictive of the disease severity. This point is not discussed. However, because mutations are localized on the P loop it could be that the selectivity of the channel to K+ is reduced by the variants and that the channels become permeable to Na+. This could eventually explain why the global current density was slightly affected. This was particularly the case in configuration mimicking patient’s situation. This could be tested simply first by measuring the apparent reversal potential of the current (from the inward current relaxation following the hyperpolarization of the cells after the depolarizing voltage steps) and then, if reversal potential is more depolarized,  by  the substitution of  extracellular Na+ with choline.  A loss of K+ selectivity has been described in for the KCNT2 subunit of Kna Channels related to epileptic encephalopathy (see Gururaj, S., et al (2017). A de novo mutation in the sodium-activated potassium channel KCNT2 alters ion selectivity and causes epileptic encephalopathy. Cell Rep. 21, 926–933). This possibility need to be explored here.

2) Regarding the amplitude of the currents shown in Fig 1 and 2 Aa,b,c or B.  The maximum amplitude measured at +40 mV is around 1000 pA for KCNQ2 and 3000 pA for heteromeric channels.  It is difficult to understand how current density (pA/pF) reach so high values in histograms depicted under the traces. It looks like that authors forgot to divide current amplitude by the capacitance of the cells. This should be corrected

3) Fig.4: Could authors be more precise regarding the predication that the diameter of the pore of mutant channels is different from that of wild type channels. No values are indicated

4) There is lot of repetition regarding the measures of the current density. Values are indicated in the histograms, table 1, and supplementary table 1. Same for G/Gmax . Moreover it is not indicated at which membrane potential correspond the mean current densities indicated in table 1.

5) Graphs and traces corresponding to the effect of the p.Pro285Thr are missing.

6) Quantification of Western blot is not shown. Moreover the gel with quantifications (when done) should be showed in a separate figure.

7) Number of cells recorded for this study is not sufficient (8 to 10 cells for heteromeric conditions). Moreover for homomeric conditions, authors should also increase the number of mutant cells recorded for the comparison (22 WT vs only  8-10 mutant cells).

Others remarks

Methods :  2.6 Whole cell patch-clamp analysis

The sentence “The measurements of the voltage dependence of  activation….. was repeated twice.  In first sentence voltage steps were performed from holding potential of 0 mV. In the second sentence the holding potential was–80 mV.

Figures

The size of the figures is very small and it is not indicated what the X and Y axes correspond to (sec and pA respectively). In addition, the authors should show the voltage steps below the traces.

Author Response

Feb. 10 (cells-1445147)

Reviewer 2

In the manuscript untitled “KCNQ2 selectivity filter mutations cause Kv7.2 M current dysfunction and configuration changes manifesting as epileptic encephalopathies and autistic spectrum disorders” , Lee and Collaborators describe the clinical presentation  of three patients harboring the variants p.Thr287Ile, p.Gly281Glu, p.Pro285Thr of the Kv7.2 subunit and the functional consequence of these variants  on M current carried by homomeric Kv7.2 channels and by heteromeric Kv7.3 and Kv7.2 channels in configuration that mimic patient’s situation. In addition authors performed computational protein analysis of the mutations. The three patients harboring these mutations exhibited developmental delay including dyskinetic movement disorders and ASD . The goal of this study was to see if there is any phenotype-genotype correlation between the functional impact of the variant and the disease severity. Although such type analysis is not new,  this study describe the functional impact of three new variants in KCNQ2-EE.  I have some concerns regarding this study.

Reply: We are grateful for the opportunity to improve our manuscript and we thank the editorial board and the reviewers for their thoughtful and helpful comments and criticisms. We have modified the paper as suggested. Following are our point-by-point responses. We have also highlighted the principal changes in the “Revised Text”.

1)Important informations are missing in the electrophysiological study: Kinetics of activation and deactivation and reversal potential of the current are not shown. In addition the effect of the 3 mutations on current density is not so strong (< 20% of decrease). This decrease is similar to what is observed in BFNS. Therefore the amount of current decrease is not predictive of the disease severity. This point is not discussed. However, because mutations are localized on the P loop it could be that the selectivity of the channel to K+ is reduced by the variants and that the channels become permeable to Na+. This could eventually explain why the global current density was slightly affected. This was particularly the case in configuration mimicking patient’s situation. This could be tested simply first by measuring the apparent reversal potential of the current (from the inward current relaxation following the hyperpolarization of the cells after the depolarizing voltage steps) and then, if reversal potential is more depolarized,  by  the substitution of  extracellular Na+with choline.  A loss of K+ selectivity has been described in for the KCNT2 subunit of Kna Channels related to epileptic encephalopathy (see Gururaj, S., et al (2017). A de novo mutation in the sodium-activated potassium channel KCNT2 alters ion selectivity and causes epileptic encephalopathy. Cell Rep. 21, 926–933) This possibility need to be explored here. (showed the tail analysis in supplementary file?)

Reply: We presumed the reviewer asked to analyze the tail currents in the variants. We agreed and appreciate the reviewer’ opinions. We have analyzed the tail currents in the mutants. The new Figure for tail analysis exhibited in the Supplementary Figure 2.

Supplementary Figure 2. Tail currents in wild type and mutations are shown. (A) The current (I/I max) at +40 mV potential showed the currents was lowest in homomeric p.Gly281Glu, then p.Thr287Ile and KCNQ2 WT, in that order. However, the curve of the p.Gly281Glu and p.Thr287Ile cells were increased in the KCNQ2 WT + p.Gly281Glu and in the KCNQ2 WT + p.Thr287Ile. (B) In heteromeric Kv7.2 + Kv7.3 and Kv7.2 + Kv7.3 + mutants, the curves in p.Gly281Glu and p.Thr287Ile were still lower than in the WT cells. S, second.”

In Result, Conductance–current curves in p.Thr287Ile, p.Gly281Glu, p.Pro285Thr and KCNQ2 WT, paragrah 3, we added:

Tail currents in the WT and mutations are shown (Supplementary Figure 2). The current (I/I max) at +40 mV potential showed that currents were the lowest in homomeric p.Gly281Glu, then in p.Thr287Ile, and in KCNQ2 WT. However, the curve of the p.Gly281Glu and p.Thr287Ile cells increased in the KCNQ2 WT + p.Gly281Glu and in the KCNQ2 WT + p.Thr287Ile (Supplementary Figure 2A). In heteromeric Kv7.2 + Kv7.3 and Kv7.2 + Kv7.3 + mutants, the p.Gly281Glu and p.Thr287Ile curves were lower than in those in the WT cells (Supplementary Figure 2B), with currents decaying quicker than those in the WT.

In Discussion, paragraph 5, line 5-14, we added

“Changes in the functional current of the KCNQ2 mutants were not necessarily correlated to the phenotype. As these mutations were localized on the P loop, the selectivity of the channel to K+ was reduced by variants, and channels became permeable to Na+. This could ultimately explain why global currents were slightly affected. The loss of function is the major mechanism in KCNQ2 EE with de novo mutations. “Change of function” has been reported (Gururaj et al.,2017) recently in a KCNT2 de novo mutation causing EE. There is also an alternative mechanism particularly for mutations located in the SF region.”

  We added a new reference 19.

  1. Gururaj, X, Palmer E.E., Sheehan, G.D., Kandula, T., Macintosh, R., … Bhattacharjee, A. (2017). A De Novo Mutation in the Sodium-Activated Potassium Channel KCNT2 Alters Ion Selectivity and Causes Epileptic Encephalopathy. Cell Rep, 21(4):926-933. doi: 10.1016/j.celrep.2017.09.088.

2) Regarding the amplitude of the currents shown in Fig 1 and 2 Aa,b,c or B.  The maximum amplitude measured at +40 mV is around 1000 pA for KCNQ2 and 3000 pA for heteromeric channels.  It is difficult to understand how current density (pA/pF) reach so high values in histograms depicted under the traces. It looks like that authors forgot to divide current amplitude by the capacitance of the cells. This should be corrected

Reply: Thanks for the valuable opinion. We have corrected this problem and changed to express the conduction curve in Figures and text by currents amplitude (pA).

3) Fig. 4: Could authors be more precise regarding the predication that the diameter of the pore of mutant channels is different from that of wild type channels. No values are indicated

Reply: In Figure 7 (old Figure 4) legend, line 11-17, we have mentioned:

The diameters of the KCNQ2 channel pores were determined by calculating the distance from protein 258 to 285 (A), 285 to 295 (B), 295 to 276 (C), and 276 to 258 (D) The pore configuration was expressed as A × B × C × D. The mutations change the configurations of pores in p.Thr287Ile (from 31,617.3 in WT changed to 27,673.6), in p.Gly281Glu (from 31,617.3 changed to 27,673.6 and p.Pro285Thr (from 31,617.3 to 27,673.6)

In Figure 7, legend, line 21-27, we have mentioned:

The diameter of the KCNQ2 channel pores were determined by calculating the distance from protein 258 to 276 (A1), 276 to 295 (B1), 241 to 276 (C1), and 241 to 295 (D1). The pore distance was expressed as A1 × B1 × C1 × D1. Mutations changed the configurations of pores in p.Thr287Ile (from 163,809.4 in the WT to 78,808.7), in p.Gly281Glu (not different from 163,809.4 in WT ), and p.Pro285Thr (from 163,809.4 in the WT to 77,691.0).

 In Figure 7 (old Figure 4), changed to:

Figure 7. Schematic representation of the KCNQ2 subunit with the position of the mutations of p.Thr287Ile. The predicted 3D model of the KCNQ2 channel protein (c5vmsA_.1.pdb) was then used to analyze the structural differences between KCNQ2 WT and mutations by using Swiss-PdbViewer and PyMOL, respectively. Mutation (p.Thr287Ile, p.Gly281Glu and p.Pro285Thr) sites at the selectivity filter (SF)) might alter accessibility for potassium ions through the channels. The p.Thr287Ile p.Gly281Glu and p.Pro285Thr are located in the SF of the pore domain and cause the KCNQ2 protein configuration of pore domain change according to the molecular model. (A) The pore change were determined by calculating the distances from Asn (p.258) to Pro (p.285), Pro (p.285) to Ala (p.295), Ala (p.295) to Thr (p. 276), and Thr (p.276) to Asn (p.258) respectively. The diameters of the KCNQ2 channel pores were determined by calculating the distance from protein 258 to 285 (A), 285 to 295 (B), 295 to 276 (C), and 276 to 258 (D) The pore configuration was expressed as A × B × C × D. The mutations change the configurations of pores in p.Thr287Ile (from 31,617.3 in WT changed to 27,673.6), in p.Gly281Glu (from 31,617.3 changed to 27,673.6 and p.Pro285Thr (from 31,617.3 to 27,673.6) (B) (a) The pore change were also determined by calculating the distances from Asn (p.258) to Thr (p.276), Thr (p.276) to Ile (p.241), Ile (p.241) to Ala (p.295), and Ala (p.295) to Asn (p.258) respectively. (b) The pore configurations were changed by the mutations of p.Thr287Ile and p.Pro285Thr. Mutations changed configurations of pores in p.Thr287Ile, p.Gly281Glu, and p.Pro285Thr. The diameter of the KCNQ2 channel pores were determined by calculating the distance from protein 258 to 276 (A1), 276 to 295 (B1), 241 to 276 (C1), and 241 to 295 (D1). The pore distance was expressed as A1 × B1 × C1 × D1. Mutations changed the configurations of pores in p.Thr287Ile (from 163,809.4 in the WT to 78,808.7), in p.Gly281Glu (not different from 163,809.4 in WT ), and p.Pro285Thr (from 163,809.4 in the WT to 77,691.0).

4) There is lot of repetition regarding the measures of the current density. Values are indicated in the histograms, table 1, and supplementary table 1. Same for G/Gmax . Moreover it is not indicated at which membrane potential correspond the mean current densities indicated in table 1.

Reply: 1. Thanks for the opinion. We believed the reviewer is mentioned about Table 2 (not Table 1) for the membranous potential corresponding in currents. Table 2 exhibited the average currents (pA) from -80 to +40 mV in the transfected cell. 2. We have shortened the Result section.

In Table 2, footnote, changed to:

WT, wild type; V is the test potential; V½, half-maximal activation voltage; SD, standard deviation.The data were then fit to a Boltzmann distribution of the following form: G/Gmax = 1/(1 + exp[(V-)/dx]). Current was expressed as average currents (pA) from -80 to +40 mV in the transfected cells. * indicates P < 0.05 compared with KCNQ2 WT; **, P < 0.005 compared with KCNQ2 WT. Bold font indicates significantly different from WT. Data rounded off to the first decimal place. Homomeric transfected variants and heteromeric transfected KCNQ2 WT + variants were compared with the current in KCNQ2 WT (2 ug) respectively. Heteromeric KCNQ2 WT + KCNQ3 WT + variants were compared with the current in KCNQ2 WT + KCNQ3 WT (1ug:1 ug) respectively.

5) Graphs and traces corresponding to the effect of the p.Pro285Thr are missing.

Reply: We exhibited the p.Pro285Thr conduction curve in the new Figure 4.

In Figure 4, legends, we added:

Figure 4. (A) Cells transfected with p.Pro285Thr exhibited lower [p < .05 (p values see Supplementary Table 1); -10 to +40 mV ) currents in homomeric p.Pro285Thr and in heteromeric p.Pro285Thr + KCNQ2 WT [p < .05 (p values see Supplementary Table 1); +10 to +40 mV ] compared with KCNQ2 WT correspondingly.* homomeric p. Gly281Glu versus KCNQ2 WT; # p. Gly281Glu + KCNQ2 WT versus KCNQ2 WT. (B) The KCNQ2 WT + p.Pro285Thr + KCNQ3 WT exhibited lower current densities [p < .05 (p values see Supplementary Table 1) compared with KCNQ2 WT + KCNQ3 WT from +10 to +40 mV conditional stimulation.

6)Quantification of Western blot is not shown. Moreover the gel with quantifications (when done) should be showed in a separate figure.

Reply: We added the new Figure 6 to demonstrate quantification of western blot.

Figure 6. Western blotting demonstrated the protein expression in KCNQ2 WT, p.Thr287Ile, and p.Gly281Glu. KCNQ2 protein expression was not significantly (N = 3)

7) Number of cells recorded for this study is not sufficient (8 to 10 cells for heteromeric conditions). Moreover for homomeric conditions, authors should also increase the number of mutant cells recorded for the comparison (22 WT vs only  8-10 mutant cells).

 Reply: Regarding the numbers of mutant cell, the numbers of cell are similar to other studies (Miceli et al., 2013; Gomis-Perez et al., 2019; Sands et al., 2019;). We discuss in the limitations.

In Discussion, last paragraph,

changed to:

This study has a few limitations. Because of the complexity of three dimensional graphics for the pore region, the predicted 3D model exhibited the pore region change by mutations. The determination of the phenotype is complex, and the phenotype may be due to the electrical charge of mutated proteins, the hydrophobic or hydrophilic characters of mutated proteins, modified genes, or acquired brain injury due to uncontrolled seizures. However, we found that the weight of mutated proteins can be a critical factor in mutations of the KCNQ2 pore region. The predicted 3D structure denotes the effect of the mutation of the protein. Using HEK293 cells as a functional study in vitro with a potassium channel and the limitations of numbers of cells might contribute to the bias in currents, however, the numbers of cell are similar to other studies (Miceli et al., 2013; Gomis-Perez et al., 2019; Sands et al., 2019;). The nonexpression of potassium ion channels in HEK293 cells makes them an excellent model for whole-cell patch-clamp studies because only minor interfering currents occur. Because of this, HEK293 cells have been widely used in cell biology, and the gene expression of HEK293 cells is similar to the gene expression of neurons.

Others remarks

Methods :  2.6 Whole cell patch-clamp analysis

  • The sentence “The measurements of the voltage dependence of  activation….. was repeated twice.  In first sentence voltage steps were performed from holding potential of 0 mV. In the second sentence the holding potential was–80 mV.

Reply: In Method,

Whole-cell Patch-Clamp Analysis,

changed to:              

For electrophysiological analysis, HEK293 cells were washed in modified Tyrode’s solution containing 125 mM NaCl, 5.4 mM KCl, 1.8 mM CaCl2, 1 mM MgCl2, 6 mM glucose, and 6 mM HEPES (pH 7.4). Patch pipettes had a resistance of 3-4 Ω when filled with a solution containing 125 mM potassium gluconate, 10 mM KCl, 5 mM HEPES, 5 mM EGTA, 2 mM MgCl2, 0.6 mM CaCl2, and 4 mM adenosine 5′-triphosphate disodium salt hydrate (Na2ATP; pH 7.2).

KCNQ2 mutations were created using a QuickChange kit (Stratagene, La Jolla, CA, USA) and verified through sequencing (Volkers et al., 2009). To measure the voltage dependence of activation, the cells were clamped using 3-s conditioning voltage pulses to potentials between −80 mV and +40 mV in 10-mV increments from a holding potential of −80 mV. Data acquisition and analysis were using analysis software (Clampex 10.0; Molecular Devices, Sunnyvale, CA, USA). The data were then fitted to a Boltzmann distribution of the following form: G/Gmax = 1/(1 + exp[(V-)/dx]). Cell capacitance was obtained by reading the settings for the whole-cell input capacitance neutralization directly from the amplifier (Selyanko, Hadley, & Brown, 2001). KCNQ2 mutation variants and KCNQ2 WT were transfected into HEK293 cells to determine the functional changes resulting in conductancecurrent curve changes (Miceli et al., 2013; Stefani, Toro, Perozo, & Bezanilla, 1994).

  • Figures

The size of the figures is very small and it is not indicated what the X and Y axes correspond to (sec and pA respectively). In addition, the authors should show the voltage steps below the traces.

Reply: We have redone the Figures. We have added the voltage-clamp steps in Figure Legends.

In Figure 2 A and Figure 3A, we have added the X and Y axes with second and pA.

In Figure 2 and 3, legends, we have added “The voltage-clamp steps were from −80 mV and +40 mV in 10-mV increments.”

Round 2

Reviewer 1 Report

Please, check for the following points:

Page 5, paragraph 2, line 188 and 189 : were or where or was? Please , decide

Page 5, line 194: ..and complete shock cell homogenate………: what does this mean.

Is it the same as  the cell mixture  in line 6 and 7?

Page 6: line 230,231: What is the sense of this clause?. Were the mutations analyzed or predicted? Which role play functional currents and structural changes?

Page 14, legend of figure 7, lines 447 and 454: …the pore change was also determined.. instead of …..the pore change were also determined….

Page 15, line 458: The diameter ……..was determined …….instead of …….The diameter……..were determined..

Please check also for other exchanges of singular vs plural verbs.

page 17, discussion; line 519:  mimicking …………instead of………… mimicked?

Page 19,List of abbreviations, line 619: EE is not explained

Page 20, line 643: informed consent statement: …….from parents of all patients

Author Response

Reviewer 1

We are grateful for the opportunity to improve our manuscript and we thank the editorial board and the reviewers for their helpful comments and criticisms. We have modified the paper as suggested. Following are our point-by-point responses. We have also highlighted the principal changes in the “Revised Text”

Please, check for the following points:

  1. Page 5, paragraph 2, line 188 and 189 : were or where or was? Please , decide

Page 5, line 194: ..and complete shock cell homogenate………: what does this mean.

Reply: We have corrected them accordingly.

Is it the same as  the cell mixture  in line 6 and 7?

Reply: Yes, it is the same as the cell mixture in line 6 and 7.

  1. Page 6: line 230,231: What is the sense of this clause?. Were the mutations analyzed or predicted? Which role play functional currents and structural changes?

 Reply: The mutations in KCNQ2 selectivity filter were presumed to cause functional current change in Kv7.2 and configuration change of KCNQ2 protein.

 In RESULTS, first paragraph, changed to:  

Three patients with mutations in SF presented as KCNQ2 EE and ASD. To determine that three KCNQ2 variants in the SF  cause functional current changes in HEK293 cells and to predict the consequences of mutated proteins in the KCNQ2 SF, we analyzed the functional currents in the mutations in the SF, and  predicted the structural changes at the molecular level by computational protein analysis.

  1. Page 14, legend of figure 7, lines 447 and 454: …the pore change was also determined.. instead of…..the pore change were also determined…

Page 15, line 458: The diameter ……..was determined …….instead of …….The diameter……..were determined..

. Reply: We have corrected them accordingly.

  1. Please check also for other exchanges of singular vs plural verbs.

Reply: We have used the proofreading and editing services of professional native English speaking technical editors for both the first submission and this revision.

  1. page 17, discussion; line 519:  mimicking …………instead of………… mimicked?

Reply: We have corrected them accordingly.

  1. Page 19,List of abbreviations, line 619: EE is not explained

Reply: We have corrected it.

  1. Page 20, line 643: informed consent statement: …….from parents of all patients

Reply: We have corrected it.

Reviewer 2 Report

The paper has clearly be improved however authors did not respond to one of my main concern which was the measurement of the apparent reversal potential of the current.

I am also not completly satisfied by the answer concerning the low number of cells recorded. Lot of recordings can be done using heterologous cells, There are not any limitations using HEK cells for elctrophysiological recordings 

other point: Western blot and quantification should be in a specific figure.

Author Response

Reviewer 2

We are grateful for the opportunity to improve our manuscript and we thank the editorial board and the reviewers for their helpful comments and criticisms. We have modified the paper as suggested. Following are our point-by-point responses. We have also highlighted the principal changes in the “Revised Text”

  1. The paper has clearly be improved however authors did not respond to one of my main concern which was the measurement of the apparent reversal potential of the current.

Reply: We have analyzed the tail currents (reversal potential), and found in the homomeric variants, and heteromeric variants after adding KCNQ2 and KCNQ3, the tail currents were lower than wild type correspondingly. We showed that in the new Figure 6.

  • We have made a new Figure 6.
  • In Result, Conductance–current curves in Thr287Ile, p.Gly281Glu, p.Pro285Thr and KCNQ2 WT, paragrah 3, changed to:

Tail currents in the WT and mutations are shown (Figure 6, ABC). The currents (pA) at +40 mV potential showed that currents were lower in homomeric p.Gly281Glu (513.2 ± 64.7; p = .022) and in p.Thr287Ile (499.7 ± 34.6; p = .007) than the currents in KCNQ2 WT (625.6 ± 58.1) (Figure 6, D ). However, the tail currents of the p.Gly281Glu and p.Thr287Ile cells were increased in the KCNQ2 WT + p.Gly281Glu (656.5 ± 49.4) and in the KCNQ2 WT + p.Thr287Ile (646.6. ±87.65) at +40 mV conditional voltage. In the heteromeric Kv7.2 + Kv7.3 and Kv7.2 + Kv7.3 + mutants, the currents were increased in p.Gly281Glu (1078.5 ± 153.8; p = .010) and in p.Thr287Ile (1111.6 ± 169.6; p = .033), and were still lower than in those in the KCNQ2 WT + KCNQ3 WT cells (1286.7. ±112.0) (Figure 6, D).

  • In Figure 6 (new), legend.

Figure 6. (A) Tail currents in the WT and mutations are shown at +40 mV conditional voltage. (a) KCNQ2 WT (2 ug) (n = 22), (b) KCNQ2 WT + KCNQ3 WT (1 μg:1 μg) (n = 10). (B) (a) The homomeric p.Gly281Glu (n = 8), (b) heteromeric p.Gly281Glu + KCNQ2 WT (1ug: 1ug) (n = 8), (c) heteromeric KCNQ2 WT + p.Gly281Glu + KCNQ3 WT (0.5 μg: 0.5 ug:1 μg) (n = 8)  (C) (a) The homomeric p.Thr287Ile (2 ug) (n = 10), (b) heteromeric p.Thr287Ile + KCNQ2 WT (1ug: 1ug) (n = 10), (c) heteromeric KCNQ2 WT + p.Thr287Ile + KCNQ3 WT (0.5 μg: 0.5 ug:1 μg) (n = 10) (D) The homomeric variants had lower currrents in p.Gly281Glu (513.2 ± 64.7; p = .022) and in p.Thr287Ile (499.7 ± 34.6; p = .007) than the currents in the KCNQ2 WT (625.6 ± 58.1). In the heteromeric Kv7.2 + Kv7.3 and Kv7.2 + Kv7.3 + mutants, the currents were lower in p.Gly281Glu (1078.5 ± 153.8; p = .010) and in p.Thr287Ile (1111.6 ± 169.6; p = .033), than in those in the KCNQ2 WT + KCNQ3 WT cells (1286.7. ±112.0).

  • We have deleted the Supplementary Figure 2.
  1. I am also not completly satisfied by the answer concerning the low number of cells recorded. Lot of recordings can be done using heterologous cells, There are not any limitations using HEK cells for elctrophysiological recordings 

Reply: 1) We agreed that low number of cells can contribute a limitation. We did a comprehensive review of literature that the published papers caused the significant findings despite the number of cells were similar to the numbers in our study. We had mentioned this in the final section of Discussion.

This study has a few limitations. Because of the complexity of three dimensional graphics for the pore region, the predicted 3D model exhibited the pore region change by mutations. The determination of the phenotype is complex, and the phenotype may be due to the electrical charge of mutated proteins, the hydrophobic or hydrophilic characters of mutated proteins, modified genes, or acquired brain injury due to uncontrolled seizures. However, we found that the weight of mutated proteins can be a critical factor in mutations of the KCNQ2 pore region. The predicted 3D structure denotes the effect of the mutation of the protein. Using HEK293 cells as a functional study in vitro with a potassium channel and the limitations of numbers of cells might contribute to the bias in currents, however, the numbers of cell are similar to other studies, which cause significant findings (Abidi et al., 2015; Devaux et al., 2016; Millichap et al., 2017; Miceli et al., 2013; Gomis-Perez et al., 2019; Orhan et al., 2014; Sands et al., 2019;). The nonexpression of potassium ion channels in HEK293 cells makes them an excellent model for whole-cell patch-clamp studies because only minor interfering currents occur. Because of this, HEK293 cells have been widely used in cell biology, and the gene expression of HEK293 cells is similar to the gene expression of neurons.

  1. other point: Western blot and quantification should be in a specific figure.

Reply: We have redone the Figure 3 and made a new Figure 7 (western blot).

  • In Result, Phenotypes, KCNQ2 Protein Expression, and configuration change on Cell Membranes, changed to:

“After analyzing KCNQ2 protein expression for various variants, KCNQ2 protein  expression on cell membranes did not differ significantly (n = 3) in KCNQ2 WT, p.T287I and p.Gly281Glu (Figure 7, AB).”

  • In new Figure 7, legend:

Figure 7. (A) Western blotting demonstrated the protein expression in KCNQ2 WT, p.Thr287Ile, and p.Gly281Glu. KCNQ2 protein expression was not significantly (N = 3) (Supplementary Figure 1 ) different in p. T287I and p. Gly281Glu. Protein expression on cell membranes did not differ significantly for both mutations and KCNQ2 WT. (B) Western blotting demonstrated the protein expression in KCNQ2 WT, p.Thr287Ile, and p.Gly281Glu. KCNQ2 protein expression was not significantly (N = 3).

Round 3

Reviewer 2 Report

The authors did not respond to my concern: the reversal potential of the tail current. This is not shown. I do not care about the amplitude of the tail current at +40 mV. I was interested to know at which membrane potential tail current is null and if there are differences between the value in wild-type and mutants. It is pity that authors did not try to do such measurement. So figure 6 as it is shown has no interest and could be removed.